# Tracking country-level mitigation progress using NGHGI-consistent carbon budgets

Konstantin Weber ✉, Cyril Brunner & Reto Knutti

The remaining carbon budget (RCB) of countries provides a benchmark for evaluating national mitigation efforts and was central to a recent European Court of Human Rights' ruling. However, estimates of national RCBs are inconsistent with $CO_2$ accounting in national greenhouse gas inventories (NGHGIs). Here, we align RCBs with NGHGI accounting standards. For 2024, NGHGI alignment reduces the 1.5 °C (50%) global RCB by ~100 $GtCO_2$ ($\approx$ 50%) and the 2 °C (66%) RCB by ~200 $GtCO_2$ ($\approx$ 20%). Thus, we estimate the 1.5 °C (50%) NGHGI-consistent global RCB to be depleted by 2027. We provide NGHGI-consistent national RCBs for common allocation methods and most countries. Following Paris Agreement equity principles, we find that by 2025, 64–85 countries could have exceeded their fair-share RCB for 1.5 °C (50%). While national RCBs depend on normative choices and are unlikely to directly drive negotiations, our framework enables more methodologically robust RCB calculations to track country-level mitigation progress.

The success of global climate targets, such as those outlined in the Paris Agreement[1], depends on the mitigation efforts of individual countries. National remaining carbon budgets (national RCBs) are a conceptual tool for informing and assessing national climate policy[2,3]. They represent a country's share[4,5] of the global RCB – the maximum net $CO_2$ emissions permissible before exceeding a specific temperature threshold with a given probability[6,7] – under certain assumptions and value judgments. Irrespective of whether RCBs drive global policy and mitigation efforts (see discussion) or simply track expected national progress, they are simple metrics to use. Once calculated, national RCBs can be compared with future $CO_2$ emissions expected from national climate targets to infer the target's compatibility with Paris Agreement temperature limits. This comparison, however, is methodologically challenging because the definition of anthropogenic $CO_2$ emissions embedded in national climate targets differs from the definition used in global RCB calculations[8,9]. This mismatch must be taken into account to ensure consistent quantification of RCBs for accurate assessments of national climate policy. Conceptually, this mismatch cannot be entirely eliminated, because the scientific foundation of the global RCB is undermined when using the reporting guidelines[9,10] used in national greenhouse gas inventories (NGHGIs), as additional global warming does not stop when reaching net zero $CO_2$ emissions[11].

Determining a country's share of the global RCB involves three steps: (1) defining the global RCB available, (2) selecting a framework for the distribution of the global RCB to all countries (i.e., the allocation principle), and (3) operationalizing and applying the allocation principle with available data (i.e., the specific allocation method). Although there exists an extensive literature on sharing the global RCB (steps 2 and 3)[4,12–20], there are common methodological inconsistencies in step one – the definition of the distributable global RCB – that are not considered (e.g.,[14,20–22]) or addressed non-transparently (e.g.,[23]).

Methodological inconsistencies arise due to differences in how anthropogenic $CO_2$ emissions are accounted for in scientific modeling conventions versus the reporting guidelines used in NGHGIs. Global RCB values in IPCC reports[7] (IPCC-based RCBs) or other scientific assessments[24,25] are derived under scientific modeling conventions. In contrast, it is the NGHGIs and their accounting conventions that form the basis for most emission reduction targets in nationally determined contributions (NDCs) and national climate strategies[22,26].

Scientific modeling conventions and NGHGI accounting differ in their definitions of anthropogenic $CO_2$ emissions in two ways: The first difference occurs in the definition of land-use, land-use change, and forestry (LULUCF) $CO_2$ emissions, where scientific modeling conventions and NGHGIs implicitly attribute different parts of atmosphere-

Institute for Atmosphere and Climate Science ETH Zurich, Zurich, Switzerland. ✉e-mail: konstantin.weber@env.ethz.ch

land $CO_2$ fluxes to human influence[11,27-29]. In scientific modeling conventions that underpin the IPCC assessments, indirect $CO_2$ fluxes – the fluxes driven by human-caused changes to the environment, such as elevated atmospheric $CO_2$ levels, higher temperatures, and changes in nutrient supply – are not counted as anthropogenic $CO_2$ fluxes, but are considered to be part of the natural land sink[11]. These indirect effects, also termed "passive", have so far led to a strong net uptake of atmospheric $CO_2$[30]. NGHGI accounting[9,10], however, relies largely on observational data, making it typically difficult to fully separate passive $CO_2$ fluxes from $CO_2$ fluxes due to direct anthropogenic influence (e.g., de-, re-, and afforestation or forest management). For practical reasons, NGHGI accounting uses land that is classified by countries as "managed" as an indicator for the land where anthropogenic $CO_2$ fluxes occur[27,28]. Part of what is considered the natural $CO_2$ sink by modelers is incidentally included as an anthropogenic $CO_2$ sink in NGHGIs, as countries largest in area tend to classify most (if not all) land as managed, and isolation of passive $CO_2$ fluxes remains imperfect in NGHGI accounting[29]. This results in a fundamental mismatch of 5–7 $GtCO_2$ per year, corresponding to 12–17% of 2023's global total anthropogenic $CO_2$ emissions[30,31]. This gap continues to evolve depending on the climate scenario[8,32] and the land classification by countries. This difference in methodology also explains why IPCC reports[7] and assessments of the Global Carbon Project[30] find that the LULUCF sector is a net source of $CO_2$ emissions, while NGHGIs consistently report it as a net sink[8,27-29,32].

Crucially, under scientific modeling conventions, the concept of reaching net zero $CO_2$ to stop global temperature rise works just because passive $CO_2$ fluxes are classified to be natural (non-anthropogenic)[11]. After reaching net zero $CO_2$ emissions, continued passive $CO_2$ uptake by the land and oceans leads to decreasing atmospheric $CO_2$ concentrations that are necessary to counterbalance continued warming after forcing stabilization[33]. On the contrary, under NGHGI accounting of anthropogenic LULUCF $CO_2$ emissions, achieving net zero anthropogenic $CO_2$ emissions does not halt global temperature rise, as demonstrated by Allen et al.[11]. Most of the passive $CO_2$ uptake occurring on managed land is regarded as anthropogenic $CO_2$ removal in NGHGIs[29] and can therefore be potentially used to compensate for anthropogenic $CO_2$ emissions[11]. Thus, after reaching global net zero $CO_2$ from a NGHGI accounting perspective, the IPCC-based RCB continues to deplete. Limiting warming to a set temperature threshold requires continuous net negative $CO_2$ emissions in NGHGI accounting – this makes the concept of a RCB fundamentally incompatible with NGHGI accounting. Correcting the size of the available global RCB to fit NGHGI accounting conventions therefore requires assumptions about future emission pathways[8].

The second difference relates to $CO_2$ emissions from international aviation and shipping (bunker fuels). Under the United Nations Framework Convention on Climate Change (UNFCCC), their mitigation is coordinated in cooperation with the International Civil Aviation Organization (ICAO) and the International Maritime Organization (IMO)[34]. NGHGIs still report $CO_2$ emissions from bunker fuels but exclude them from national totals[9,35]. Similarly, bunker fuel emissions are most often excluded from national climate targets. Exceptions include intra-EU aviation emissions[36], Switzerland's net zero target[37], and the UK climate target[38].

A meaningful assessment of national climate targets and ambitions requires national RCBs derived from a global RCB that is aligned with NGHGI accounting conventions. This fact is, however, usually not explicitly considered or not communicated (e.g.,[17,18,39-41]). Political and societal implications of this shortcoming arose in the ruling of the European Court of Human Rights (ECtHR) in the case of *Verein KlimaSeniorinnen Schweiz and Others v. Switzerland*[42]. The (lacking) quantification of a national RCB represented a central point in the court's argumentation (e.g., paragraphs 322–325, 360, 569–573[42]), but awareness of raised methodological mismatches was missing.

Determining a country's share of the global RCB remains a delicate task because perceived emission reduction responsibilities are not purely scientific but involve political and ethical dimensions[3,43,44]. While the Paris Agreement sets global temperature limits, the contributions of individual countries remain voluntary, submitted as NDCs[22] to the UNFCCC. The UNFCCC and the Paris Agreement establish guiding principles for equity[1,45,46], which can act as guardrails for translating a global RCB into national RCBs.

In this study, we propose a correction of the distributable global RCB that improves the consistency with NGHGI accounting, and thereby strengthens the robustness of national RCBs regardless of the chosen allocation principle when used to evaluate country-level mitigation progress. To simplify such a procedure, we provide a dataset of NGHGI-consistent national RCBs for a wide range of allocation methods and for all 197 Parties to the UNFCCC. Previous studies have sometimes excluded $CO_2$ emissions from the LULUCF sector and bunker fuels when allocating emissions or carbon budgets[12,46,47], and accordingly made adjustments to distributable emissions. However, these studies did not address the conceptual mismatch between scientific modeling conventions and NGHGIs. While the implications of differences in the attribution of passive $CO_2$ fluxes in the LULUCF sector have been documented[8,32], and a correction has been included once before in the context of a national RCB[48], to our knowledge, the two necessary corrections have so far not been applied systematically to global and national RCBs. We aim to fill a gap in the scientific literature concerning an up-to-date, methodologically more robust quantification of national RCBs that combines global scope, temporal coverage, and a broad range of allocation methods. Here, we quantify the effect of the proposed correction and also examine the variation in the updated national RCBs that arises from different normative choices.

## Results

### The global NGHGI-consistent RCB

To convert the IPCC-based RCBs into NGHGI-consistent RCBs, we apply two independent absolute correction terms (Fig. 1): The first correction, the attribution of passive $CO_2$ fluxes in the LULUCF sector, strongly depends on the specific climate scenario (Fig. 1c) and the future amount and type of land classified as managed by countries. The methodological discrepancy between scientific modeling conventions and NGHGI accounting is projected to decline until global net zero $CO_2$ emissions (according to the definition in scientific modeling) are reached and switches sign afterward[8] (Supplementary Fig. 1). Using data from Gidden et al.[8], we estimate that in 2024, the adjustment reduces the global RCB by 63 (41–121) $GtCO_2$ for 1.5 °C (50%) and 117 (80–153) $GtCO_2$ for 2 °C (66%) scenarios. Values in parentheses are the 5th (of C1 and C3 scenarios, respectively) and 95th percentile (of C2 and C3 scenarios, respectively) of the estimated correction terms, spanning several tens of $GtCO_2$ of scenario uncertainty.

The second correction accounts for excluding bunker fuel $CO_2$ emissions in NGHGIs. This further reduces the NGHGI-consistent global RCB (Fig. 1b): For 2024, we estimate a reduction of 33 (15–43) $GtCO_2$ for 1.5 °C and 77 (27–89) $GtCO_2$ for 2 °C scenarios, respectively. Future $CO_2$ emissions from bunker fuels also depend on scenario characteristics and data source (Fig. 1b), with AR6 scenarios projecting slightly higher values than other assessments.

Consequently, we estimate a 2024 NGHGI-consistent RCB of 109 $GtCO_2$ (41–149 $GtCO_2$ considering the spread in correction terms) and 709 (660–796) $GtCO_2$ for limiting warming to 1.5 °C (50%) and 2 °C (66%), respectively (red bars in Fig. 1a, d and Supplementary Fig. 3). For 1.5 °C (50%), this corresponds to a 47% (27–80%) reduction due to NGHGI alignment, equivalent to twice the anthropogenic $CO_2$ emissions in 2024[30]. For 2 °C (66%), we find a 21% (12–27%) reduction due to NGHGI alignment in 2024. For both temperature limits, the majority of this reduction stems from the difference in the definition of

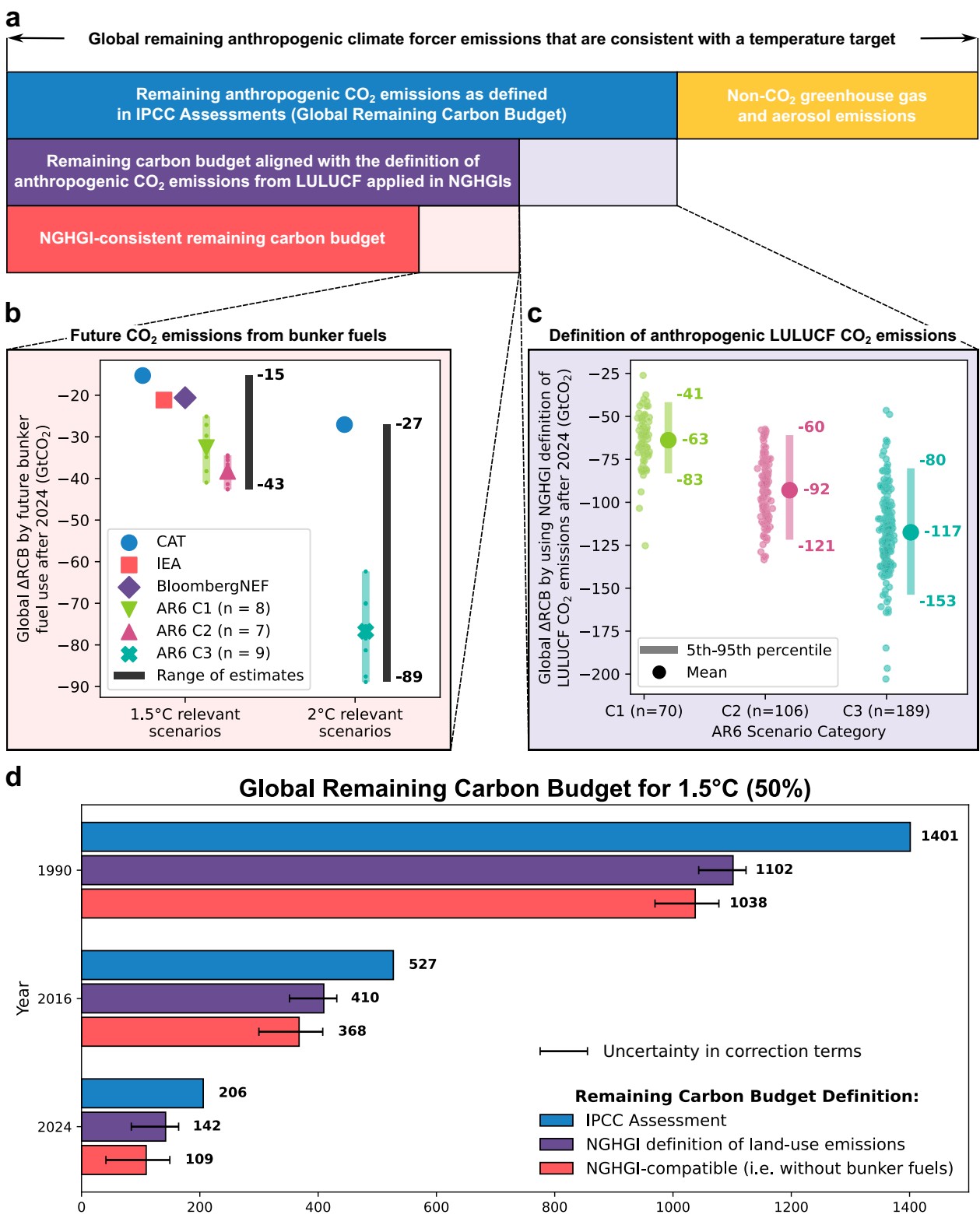

anthropogenic LULUCF $CO_2$ emissions. The absolute difference between RCB definitions increases when going back in time (Fig. 1d), because of past differences in the reported anthropogenic LULUCF $CO_2$ emissions and accumulating $CO_2$ emissions from bunker fuels. When the Paris Agreement was adopted, the 1.5 °C NGHGI-consistent global RCB was 368 $GtCO_2$ – 30% lower than the IPCC-based RCB of 527 $GtCO_2$.

## Allocation of the NGHGI-consistent budget to countries

Using NGHGI-consistent global RCBs, we derive national RCBs from 1990 to 2022/2023 for a wide range of allocation principles and countries and make this dataset of NGHGI-consistent national RCBs available (see data availability section). The Paris Agreement refers to "the principle of equity and common but differentiated responsibilities and respective capabilities, in the light of different national

**Fig. 1 | Conversion of the IPCC-based to a NGHGI-consistent global RCB.**
**a** Schematic representation of the framework suggested to convert cumulative anthropogenic $CO_2$ emissions consistent with staying below a certain temperature limit (the global RCB), as defined in IPCC reports, to a NGHGI-consistent global RCB. Two correction terms are applied to the IPCC-based global RCB: The future $CO_2$ emissions from bunker fuels (**b**) and a change in the definition of anthropogenic LULUCF $CO_2$ emissions (**c**). Bar sizes in (**a**) are indicative and not necessarily proportional to actual values. **b** The range of bunker fuel emissions in different scenarios from 1 January 2024 to the year of net zero $CO_2$. **c** The change in the 2024 global RCB by adopting the definitions of anthropogenic LULUCF $CO_2$

emissions employed in NGHGIs (obtained from Gidden et al.[8]). C1, C2, and C3 denote IPCC AR6 scenarios with different maximum temperatures in (**b**) and (**c**): C1: Below 1.5 °C (50%) with no or limited overshoot; C2: Below 1.5 °C (50%) with high overshoot; C3: Below 2 °C (66%). **d** Comparison of alternative definitions of the global 1.5 °C-compatible RCB (50%) for 1990, 2016 (around the time of Paris Agreement adoption), and 2024 (the last year global data is available for this study). Error bars reflect uncertainty in correction terms. The lower (upper) bound is derived from the maximum (minimum) of assessed future bunker fuel emissions and the 95th percentile of C2 scenarios (5th percentile of C1 scenarios) for the assessed mismatch in future LULUCF $CO_2$ emissions.

circumstances"[1]. This terminology goes back to the UNFCCC Earth Summit in Rio de Janeiro 1992[45], but does not define metrics for allocation. The Paris Agreement formulation has generally been interpreted in simple words as "all countries should contribute, but those who have emitted more in the past and have capacity (e.g., financial, technical) should contribute more".

For illustration, we consider already established allocations based on equal-per-capita (EPC), cumulative equal-per-capita (cEPC), different weighting of economic capacity (CAP)[17,18], Bretschger burden sharing (Bretschger)[15], grandfathering, (cumulative) equal-per-capita with historical responsibility[4,47] for territorial or consumption-based emissions (cEPC+Terr / EPC+Terr, or cEPC+Cons / EPC+Cons, respectively), and capacity with historical responsibility (CAPRES). Years contained in allocation labels refer to the starting year of historical responsibility or aggregation (in the case of cumulative indicators). The scaling factor $\sigma$ acts as a variable weight for economic capacity, as detailed in Equations (11)–(18) in the Methods. We do not attempt to judge the different allocation assumptions and their implications in this work, but rather to correct them for consistency with the NGHGI methodology and provide them to any potential user. Many other allocation criteria have been proposed and are possible, while here we provide a number of allocation principles commonly found in the literature.

We illustrate the diversity in allocation principles and methods by presenting results for four countries selected for their distinct economic profiles and emission trajectories: China, the USA, Switzerland, and Nigeria. Figure 2 shows their per-capita 1.5 °C-compatible RCBs for 2022 (analogous figures for other countries are provided with the dataset). Across most countries, allocated national RCBs vary considerably depending on the chosen allocation principle – particularly based on whether (and if so, since when) historical responsibility is accounted for. While alignment with NGHGI accounting substantially changes the size of the national RCB for some allocation principles – e.g., China's 2022 EPC-based RCB is reduced by 39%, or its 2022 CAPRES1990 ($\sigma$=1)-based RCB is almost completely depleted – in terms of magnitude, the choice between the different allocation principles often has a larger effect on the resulting national RCB (see Supplementary Fig. 4 and 5 for the effect of the correction on national RCBs).

The upper ten allocations in Fig. 2 (EPC to CAP1950 ($\sigma$=1)) disregard historical responsibility, distributing the NGHGI-consistent global RCB among countries without adjustments for past emissions. This always yields positive budgets (as long as the global RCB is still positive), but the size is dependent on the year the allocation is performed: Even if a country's relative share were to remain constant over time, the global RCB decreases with time (Fig. 1d). Thus, these estimates can be misleading, in particular as certain underlying principles (e.g., grandfathering and Bretschger burden sharing) are considered ethically problematic and are not aligned with UNFCCC equity principles[43,46].

Allocation methods incorporating historical responsibility (e.g., EPC+Terr, EPC+Cons, and CAPRES) can result in negative RCBs for high cumulative emitters such as the USA, Canada, or Qatar, indicating exceedance of a country's fair-share $CO_2$ emissions under the

respective allocation principle. There is no agreement on the start of historical responsibility, and previous studies define different or a range of starting years[19,47,49]. Unsurprisingly, this choice matters for the size of national RCBs – evident in Fig. 2 for China, the USA, and Switzerland. Including model-based anthropogenic LULUCF $CO_2$ emissions in calculations of historical responsibility – rather than fossil $CO_2$ emissions alone – leads to a decrease in calculated RCBs for countries recently associated with high LULUCF $CO_2$ emissions, such as those located in South America, Sub-Saharan Africa, and Southeast Asia (Supplementary Fig. 6).

Nuances in implementation details, such as the weighting of economic capacity (by the scaling factor $\sigma$, Equations (17), (18), and (23)) and the choice between territorial and consumption-based $CO_2$ emissions, further influence national RCBs. For countries like Switzerland, Singapore, or Sweden, the consumption-based RCB is substantially lower than their territorial RCB. Among the selected countries, the sensitivity to changes in capacity weighting is evident for the USA and China. China's 2022 RCB stays positive, except under strong weighting of economic capacity from 1990 onward. Meanwhile, we find that Nigeria's RCB remains positive and largely unaffected by the allocation method – here, the most important determining factor is whether historical responsibility is being considered.

In brief, the selection of countries in Fig. 2 highlights the widely recognized strong dependence of national RCBs on both normative allocation choices and implementation details[13,16,17,21,43,50]. Here we find that for 2022 these tend to dominate over the correction applied and the sensitivity to the chosen temperature target (see Supplementary Fig. 7 for comparison with Fig. 2).

## Time-dependent national RCBs and exceedance of Paris agreement temperature limits

Calculating the NGHGI-consistent global RCB back in time (Eq. (3)) allows us to examine the time evolution of national RCBs and provide insights into how national and regional contributions to carbon budget depletion have changed over the past decades. We assess how selected national and regional RCBs have evolved since 1990, focusing on five allocation principles that interpret UNFCCC equity principles and account for historical responsibility and economic capacity since 1990 (denoted as "fair-share" allocations and marked with an asterisk in Fig. 2): EPC+Terr1990, cEPC+Terr1990, EPC+Cons1990, cEPC+Cons1990, and CAPRES1990 (scale=0.5). Figure 3 shows the time series of the full range of fair-share per-capita RCBs for China, the USA, Switzerland, and Nigeria, as well as aggregated geographical regions.

National RCB depletion over time differs widely (Fig. 3a, b): The US 1.5 °C-compatible RCB turned negative already around 2000 and continues to decline faster than the global average, China's RCB depletes more rapidly than the global average since around 2010, and Switzerland's RCB spans positive to negative values depending on the chosen allocation method over a long period of time. On a global scale, there are only three out of eight geographical regions – Africa, Asia, and Central America – that have not yet exceeded their fair share of the 1.5 °C (50%) RCB in 2022, with Europe, North America, Oceania, and South America already surpassing it when the Paris Agreement was

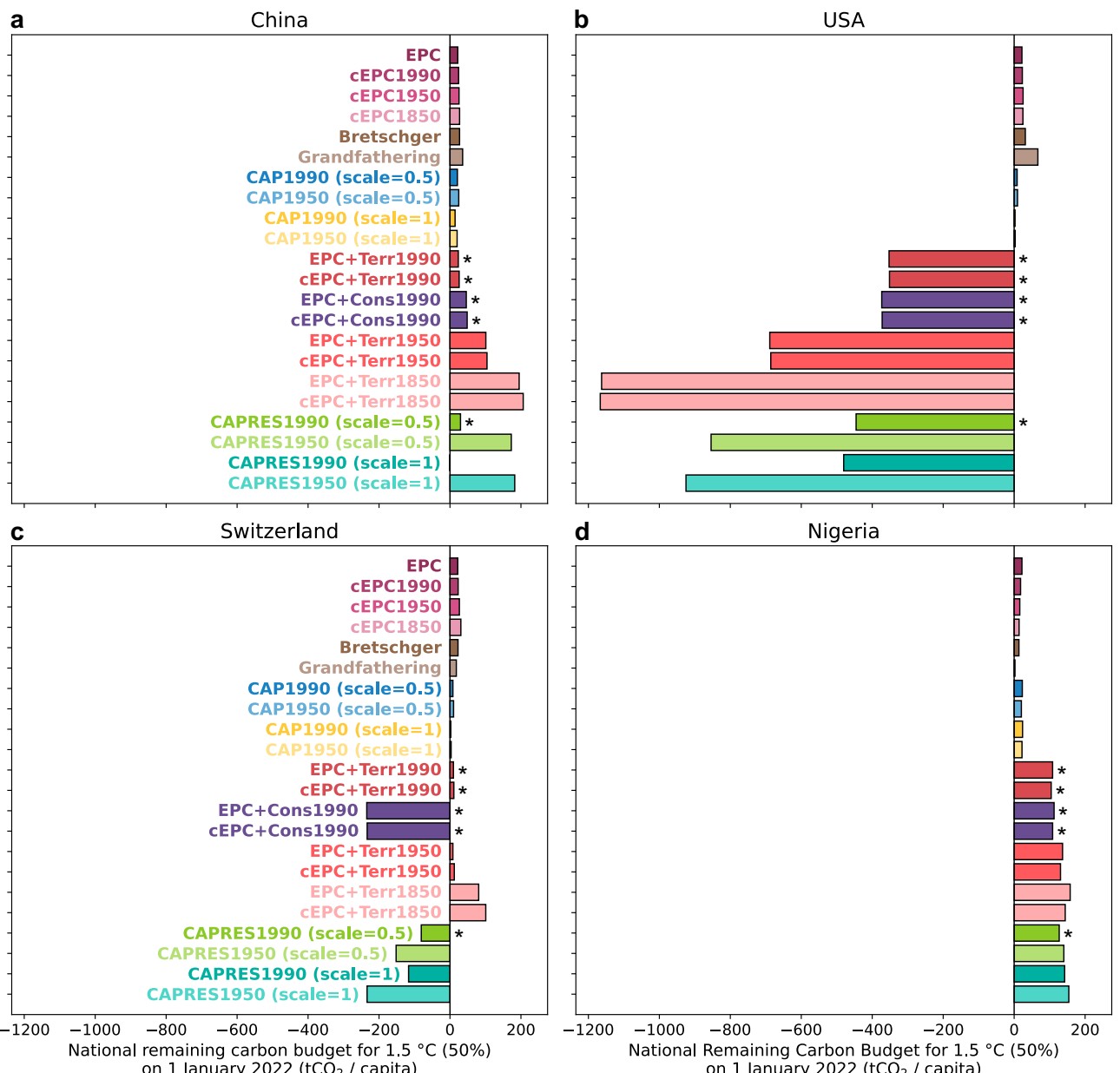

**Fig. 2 | Illustration of national RCBs from different allocation methods for four countries.** Per-capita 1.5 °C (50%) RCBs for China (**a**), the USA (**b**), Switzerland (**c**), and Nigeria (**d**) as of 1 January 2022, computed for a range of allocation methods. EPC denotes equal-per-capita allocations; Bretschger refers to Bretschger burden sharing; CAP stands for capacity-based allocations; EPC+Terr (EPC+Cons) considers historical responsibility for territorial (consumption-based) emissions, and CAPRES takes capacity and historical responsibility into account. More details on the naming of allocation methods and their implementation are found in the Methods section *Distribution of the NGHGI-consistent global RCB*. Allocation methods marked with an asterisk (*) are selected for subsequent parts of our analysis.

adopted (Fig. 3c). The depletion in per-capita RCB slows down globally and across most regions, which is an encouraging development (Fig. 3d). However, regions with lower per-capita RCBs still deplete their RCBs at a faster rate. This rate of depletion decreases over time, which is also evident in alternative country aggregations (Supplementary Fig. 8). Put simply, when measured with RCBs, every year the world is still getting more unequal, but less quickly.

On a national level, among the 197 countries that are parties to the UNFCCC[51], 57–79 countries (29–40% of all) had exceeded their fair-share RCB for limiting warming to 1.5 °C (50%) by the start of 2022 (Fig. 4). These countries collectively represent 1.6–2.3 billion people (20–29% of global population) and 46–56% of global GDP. The number of countries overshooting their fair-share RCB has been steadily rising, and extrapolated trends project this number to have reached 64–85

(32–43%) by 2025 (see map in Supplementary Fig. 9) and reach 79–97 (40–49%) by 2035. We find a similar result for 2 °C (66%): 37–57 countries out of 197 (covering 1.2–1.6 billion people and 37–47% of global GDP) had already surpassed their fair share of $CO_2$ emissions by 2022 when using NGHGI-consistent RCBs (Supplementary Fig. 10 and Supplementary Tables 1–3). Thus, many countries effectively face an accruing carbon debt[47] and the notion of a "remaining carbon budget" becomes increasingly misleading, particularly for countries in the high human development index group[52] (Supplementary Fig. 11). Without alignment to NGHGIs, exceedance shifts to later years, as indicated in gray in Fig. 4 and Supplementary Fig. 10: For example, when considering IPCC-based RCBs in 2022, the number of countries exceeding their 2 °C (66%)-compatible RCB, the population they represent and their share of global GDP are reduced to 29–48 (compared to 37–57)

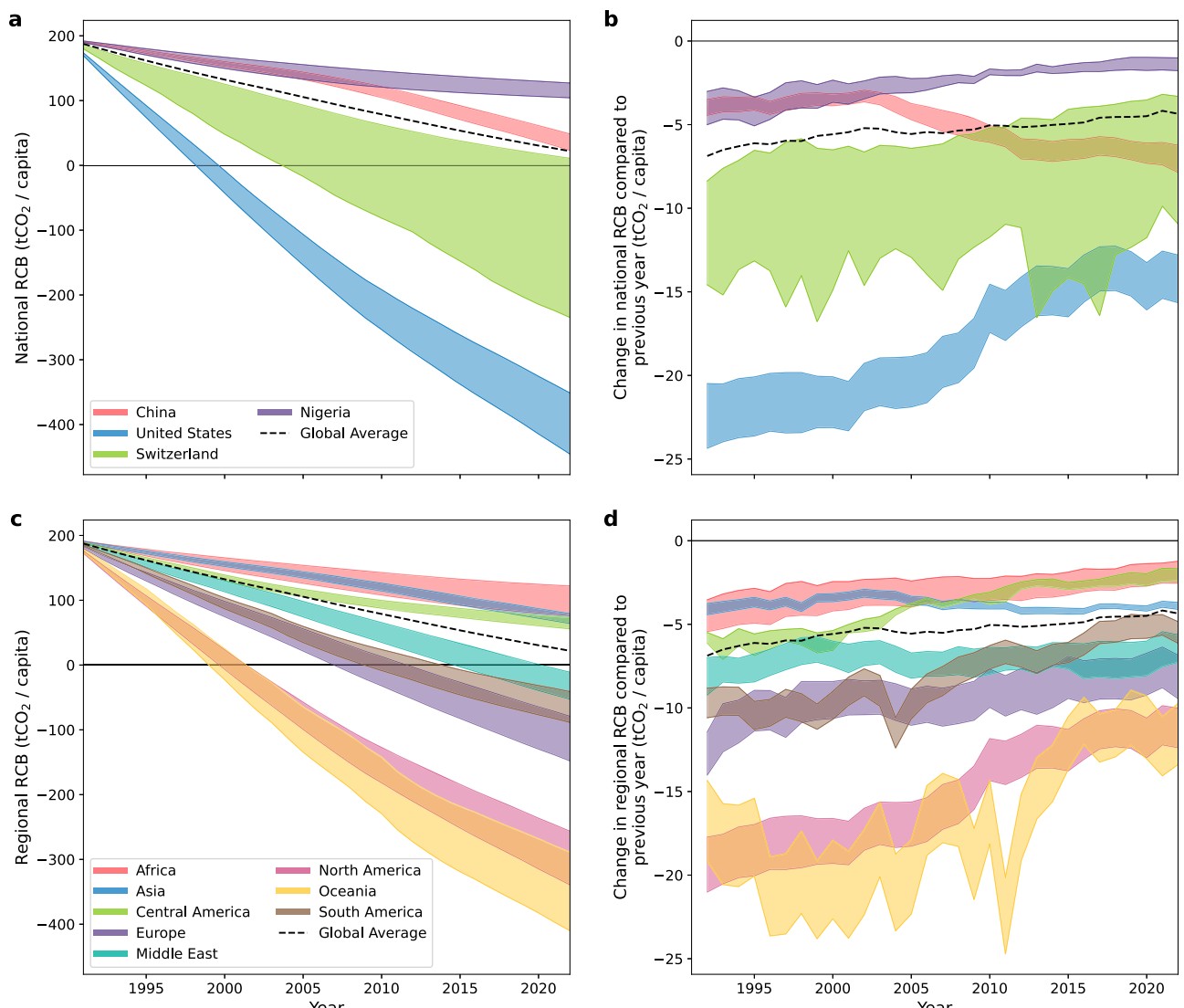

**Fig. 3 | Temporal evolution of NGHGI-consistent national and regional RCBs for 1.5 °C (50%). a** National per-capita RCBs of China, the USA, Switzerland, and Nigeria since 1990 under a selection of fair-share allocation methods. **b** Corresponding changes in national per-capita RCBs relative to the previous year. **c** Per-capita RCBs for eight geographical regions. **d** Corresponding changes in regional per-capita RCBs relative to the previous year. The colored areas indicate the full range of the five selected allocation approaches.

countries, covering 0.9–1.4 billion people, and 30–43% of global GDP, respectively.

**The case of Switzerland's RCB and NDC**

We analyze Switzerland's RCB more in depth, due to its central role in the proceedings at the European Court of Human Rights (ECtHR) in the case of *Verein KlimaSeniorinnen and Others v. Switzerland*[42], which could be a precedent for similar cases, especially after a recently published Advisory Opinion on the Obligations of States in respect of Climate Change[53] of the International Court of Justice (ICJ). In the mentioned case, Switzerland's government contended that "there was no established methodology to determine a country's carbon budget" (paragraph 570[42] and re-iterated during the proceedings of the Advisory Opinion of the ICJ[54]). Simultaneously, the court mandated that Switzerland needed to "adopt general measures specifying a target timeline for achieving carbon neutrality and the overall remaining carbon budget for the same time frame" (paragraph 550[42]). In brief, the ECtHR ultimately judged the absence of an attempt to quantify the national RCB to be a violation of Article 8 of the European Convention of Human Rights (paragraph 572[42]), which relates to the "Right to

respect for private and family life"[55]. With the framework outlined in this study, we can provide a scientifically robust estimate of Switzerland's RCB, despite the (perceived) lack of prior standardization for carbon budget calculations. We also assess Switzerland's RCB in the context of its NDC submitted in 2025[56].

We first present a revised estimate of Switzerland's 1.5 °C (67%) RCB for 2020 under the equal-per-capita (EPC) allocation principle. This allocation principle was chosen and applied in the ruling of the ECtHR and yielded a RCB estimate of 0.44 GtCO₂ for Switzerland, which was later compared to Switzerland's NGHGI-based national climate strategy (paragraph 569[42]). Using the global RCB estimate given in the IPCC AR6[7] based on scientific modeling conventions, we obtain essentially the same value of 0.44 GtCO₂ (top bar in Fig. 5). However, Switzerland's RCB reduces to 0.30 GtCO₂ when aligning the global RCB with NGHGI accounting conventions, hence, making it comparable to Switzerland's national emission reduction targets embodied in its NDC (second bar in Fig. 5). Applying the global RCB update from Lamboll et al.[24] further reduces this estimate to 0.06 GtCO₂. While these revisions do not qualitatively alter the court's argumentation, the assessment of whether Switzerland's emission

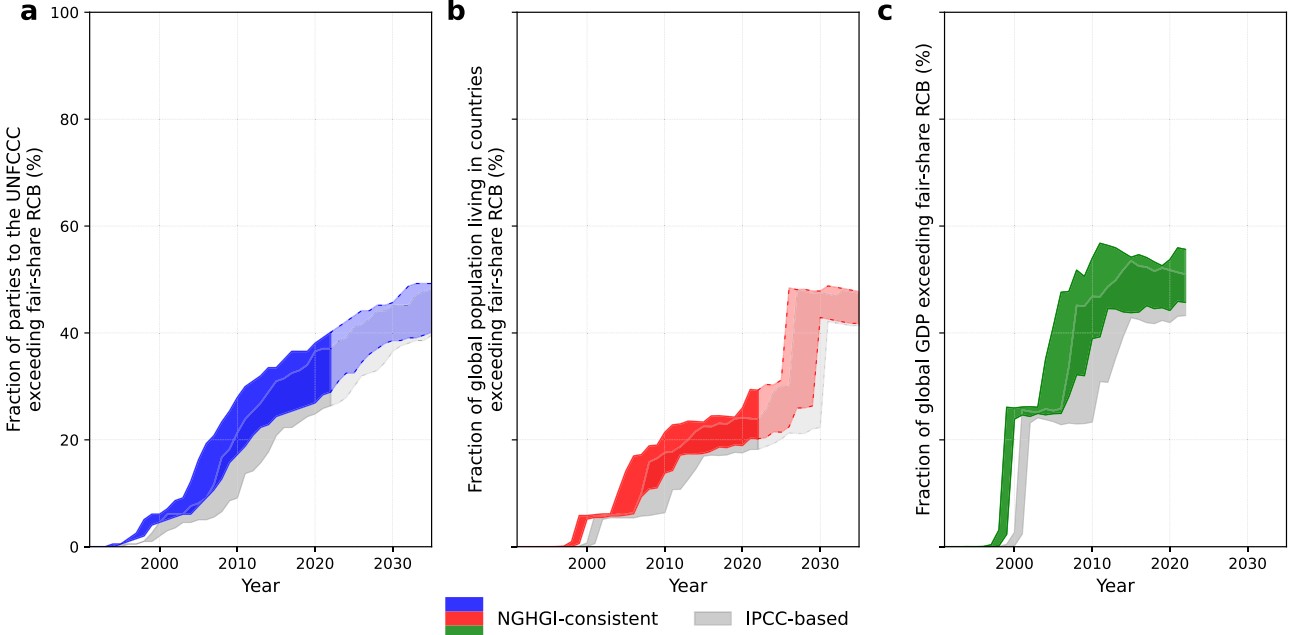

**Fig. 4 | Exceedance of national fair-share RCBs globally over time.** Fraction of UNFCCC countries exceeding their 1.5 °C (50%)-compatible fair-share RCB over time (**a**), alongside the share of global population (**b**), and GDP (**c**) they represent. The colored areas in **a**, **b**, and **c** indicate the range arising from the five selected allocation methods used to calculate national RCBs, each resulting in distinct timings of countries exceeding their fair-share of the global RCB. The gray shading corresponds to the results, if derived from an IPCC-based global RCB instead. Extrapolation to 2035 for UNFCCC parties and population is based on a quadratic extension, as described in the Methods section *Global Analysis*.

reduction targets are sufficient is changing. The originally communicated value of Switzerland's RCB (0.44 $GtCO_2$) is slightly larger than Switzerland's post-2020 cumulative $CO_2$ emissions, implied by its updated NDC[56] (vertical line in Fig. 5). Yet, both NGHGI-consistent RCB estimates are lower (by a factor of seven when considering the RCB update). Thus, Switzerland's proposed emission reductions result in higher cumulative $CO_2$ emissions than the RCB allocated to Switzerland using an equal-per-capita approach, adding detail to the court's argumentation.

For a more comprehensive assessment of Switzerland's $CO_2$ emissions reduction ambitions, we use RCBs derived from a range of allocation methods. We compare Switzerland's NGHGI-consistent 1.5 °C (50%) RCBs at the time of the Paris Agreement adoption to the cumulative $CO_2$ emissions implied by its NDC (allocation at the start of 2016). This comparison reveals that Switzerland's planned $CO_2$ emissions reductions are insufficient to remain within its fair share of the global RCB under almost all allocation methods considered (Fig. 6). The RCB exceedance extends to more than 2 $GtCO_2$ (equivalent to around 60 times Switzerland's domestic annual $CO_2$ emissions[30]) with the largest overshoots found when considering capacity or historical consumption-based emissions. The only allocation methods that provide Switzerland with a larger RCB than its NDC-consistent cumulative $CO_2$ emissions are those that consider historical responsibility for $CO_2$ emissions since 1850. This exception arises because Switzerland's historical LULUCF $CO_2$ emissions since 1850 were below the global average, leading to the buildup of a net $CO_2$ credit between 1850 and 1950. However, when historical responsibility is limited to fossil $CO_2$ emissions, Switzerland exceeds its national RCB under all assessed allocation methods (Supplementary Fig. 12). Despite the absence of one universally established allocation method, all methods considered here lead to the same conclusion: Switzerland's past and planned $CO_2$ emissions are incompatible with the global 1.5 °C limit. The magnitude of the RCB overshoot increases under allocations aligned with UNFCCC equity principles and referenced in section 4.6 of Switzerland's updated NDC[56]. When accepting a more relaxed interpretation

of the Paris Agreement of limiting warming to 2 °C (66%), Switzerland's planned contribution is sufficient under EPC allocation and inclusion of responsibility for territorial emissions, but not when considering capacity or responsibility for consumption-based emissions (Supplementary Fig. 13 and 14). Similar assessments for other countries are possible with the data we provide with this study.

## Discussion

In this study, we introduce the concept of a global NGHGI-consistent RCB for methodologically robust national RCB calculations and present a simple two-step correction framework to adjust IPCC-based RCB estimates. The RCB distributable to countries is lower than the IPCC-based RCB and not considering alignment with NGHGIs leads to a systematic overestimation of national RCBs, skewing assessments of national mitigation efforts. The necessary correction terms bear considerable uncertainty due to scenario dependence. They are likely to be underestimated (Supplementary Note 1) and the NGHGI-consistent global RCB thus still remains overestimated. Consequently, under our current assumptions, we estimate that permissible country-reported $CO_2$ emissions compatible with the 1.5 °C limit, when aggregated globally, have either already been exhausted or will reach zero by 2027 – slightly earlier than estimated in the most recent NDC synthesis report[22]. The 2024 NGHGI-consistent 2 °C (66%)-compatible global RCB amounts to approximately 700 $GtCO_2$, equivalent to just 21 years of country-reported $CO_2$ emissions from 2022[30,57], emphasizing the need for rapid global decarbonization to stay within the Paris Agreement temperature limits.

Besides uncertainties in applied correction terms, NGHGI-consistent RCB estimates inherently contain geophysical and scenario-related uncertainties from IPCC-based RCB estimates[2,6,24], particularly regarding future non-$CO_2$ emission pathways[6,58]. Less stringent non-$CO_2$ GHG mitigation could further reduce global RCB estimates by approximately 220 $GtCO_2$[7], implying depletion of the 1.5 °C (50%) NGHGI-consistent global RCB already around 2021. While integrating non-$CO_2$ GHGs into a broader global warming budget[49]

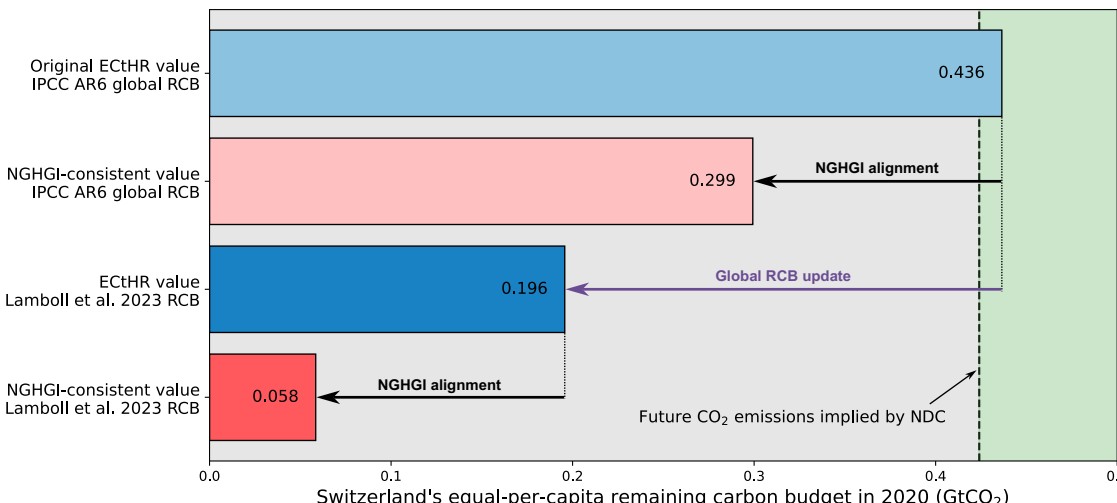

**Fig. 5 | The correction of Switzerland's equal-per-capita RCB in 2020 provided in the ruling of the ECtHR.** The original value of 0.44 GtCO$_2$ is reduced through alignment with the NGHGI accounting conventions (black arrows) and the updated global RCB estimate (purple arrow). The green area on the right indicates where Switzerland's equal-per-capita RCB is larger than the future CO$_2$ emissions implied in Switzerland's second NDC, while within the grey part, the future emissions exceed the RCB.

would offer a more comprehensive picture of climate responsibility, such an approach adds conceptual and methodological complexities. Given the conceptual simplicity, robust scientific foundation, and longstanding acceptance of carbon budgets[59–62], we focus here on CO$_2$ for introducing NGHGI-consistent RCBs.

Additional uncertainties in time-dependent NGHGI-consistent RCBs (particularly pre-2000) are due to accumulated errors in fossil CO$_2$ emissions data ($1\sigma \approx 5\%$[30]) and incompleteness of used NGHGI-reported LULUCF CO$_2$ emissions, resulting in a maximum added uncertainty of approximately 50 GtCO$_2$ ($= 1\sigma$) around 1990. National RCBs that consider historical responsibility are particularly affected by uncertainty in bookkeeping LULUCF CO$_2$ emissions[30]. By default, we include territorial LULUCF CO$_2$ emissions when calculating national carbon debts or credits (Eq. (19)–(23)). For better transparency in mitigation efforts, one might want to separate the contributions of fossil and LULUCF CO$_2$ emissions, so we provide national RCBs considering historical responsibility for only fossil CO$_2$ in the dataset, too (see Supplementary Fig. 15–17).

We see it as the role of the existing fair-share literature (e.g.,[43,46]) to inform decision-makers and courts about suitable allocation approaches. Therefore, in calculating national RCBs, we aimed to remain agnostic regarding allocation methods and provide RCBs derived from diverse equity principles. Whereas our selection is inevitably incomplete, it is sufficient to show how national RCBs are sensitive to both the underlying equity principles and their operationalization. However, this sensitivity should not obscure the need for accurate accounting and the methodological corrections we propose. Reasoning with a single allocation method paints an incomplete or misleading picture for many countries. Thus, we argue that argumentations involving national RCBs and a nuanced assessment of country-level climate targets generally require RCBs derived not only transparently and consistently but also using multiple allocation methods that are consistent with UNFCCC equity principles[1,46].

For part of our analysis, we do make an implicit value judgment in selecting five allocation methods, particularly in defining the start year of historical responsibility as 1990. This choice underestimates the carbon debt of historically high CO$_2$ emitters but reflects the choices made in previous literature[4,17–19,47] and is consistent with the time when international scientific climate change assessment and global negotiations started. Even with this conservative choice, we find that at the

time of the Paris Agreement negotiations (end of 2015), 49–69 countries had already exceeded their 1.5 °C-compatible fair share of the global NGHGI-consistent RCB – compared to 44–62 (a bias of around 10%) when using an IPCC-based RCB. This can be interpreted as an (at that time) implicit approval of a requirement for future net negative emissions, which are the primary avenue for addressing the accrued carbon debt and the pronounced asymmetry between countries with a high and those with a low to middle human development index[19,49] (Supplementary Fig. 11). Even when the global RCB for 1.5 °C or 2 °C is depleted, the quantification of national RCBs remains informative, e.g., to estimate the size of a national carbon debt and the resulting required cumulative amount of net negative CO$_2$ emissions for any internationally agreed temperature limits[47,63]. When historical responsibility is taken into account, many national RCBs are already negative today and increasingly so once the global RCB is depleted. Conversely, certain countries may keep positive budgets long after a global RCB is depleted. It is worth highlighting that under an overshoot of a global RCB, care needs to be taken when applying existing allocation methods, as interpretations of distributional approaches change when moving from a positive to a negative quantity to distribute (Supplementary Note 2). However, there is no change in the notion that countries with minimal or negative RCBs need not only to adopt and implement emission reduction strategies with the highest possible ambition[1,64], but also aim for net negative CO$_2$ emissions after they reach net zero CO$_2$ emissions.

Finally, we argue that the distribution of a global RCB to countries is not the driver of national climate policy. The concept of a global RCB has been instrumental in recognizing the need for global and country-level net zero targets. However, attempts to agree on universal allocation methods for the global RCB at the Conferences of the Parties (COPs) have not been successful, as countries are reluctant to let other entities prescribe or restrict their national policies. National RCBs inherently depend on value judgments, making agreement difficult. Deliberately selecting a single allocation method can, in an extreme case, misrepresent a country's "fair" share of the RCB. Thus, arguing with national RCBs might even prove obstructive in global negotiations. Still, assumptions around RCB calculations require harmonization. Carbon budgets (sometimes within GHG budgets) appear in policy discussions[23,65–67], the assessment of NDCs[22], and may be required to be quantified in future legal deliberations (as implied by

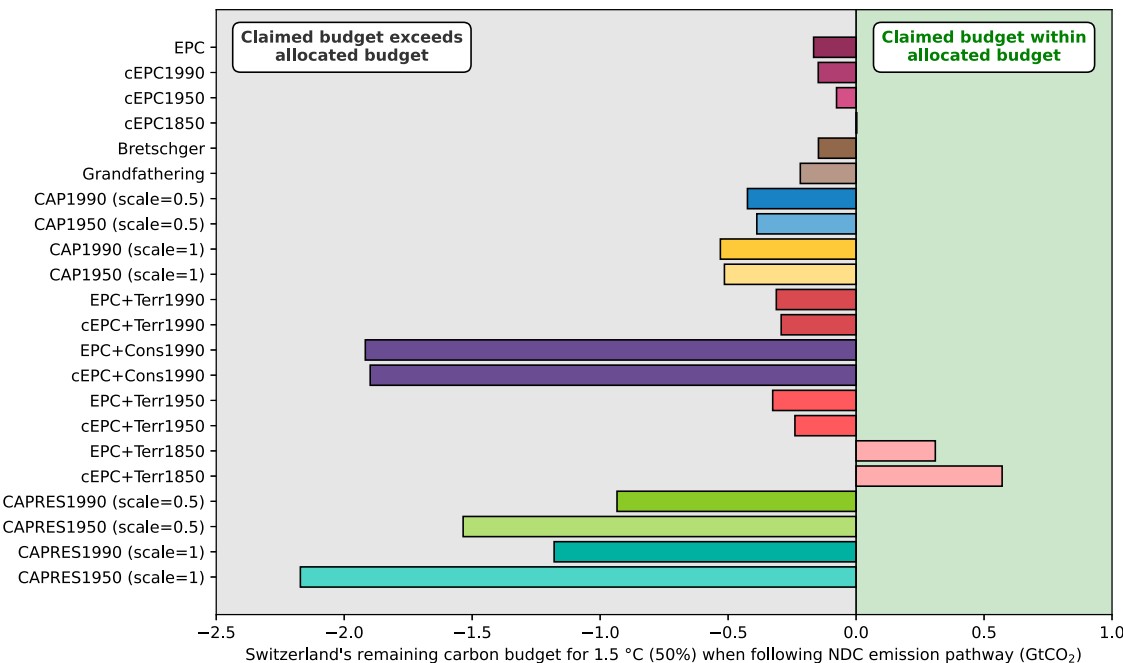

**Fig. 6 | Comparison of Switzerland's 1.5 °C (50%) RCB to Switzerland's cumulative $CO_2$ emissions implied by its second NDC.** We show the difference between Switzerland's derived RCBs – calculated by various allocation methods – and Switzerland's future $CO_2$ emissions – implied by its second NDC. RCBs are calculated for the start of 2016, which approximately coincides with the Paris Agreement adoption. Positive values suggest an allocated RCB larger than the post-2015 $CO_2$ emissions implied by Switzerland's NDC (green area to the right), while negative values indicate Switzerland overshooting its RCB (gray area to the left).

the ECtHR[42] and strengthened by the ICJ[53]). Given the cumulative nature of $CO_2$-induced warming, they are a simple and robust quantity for assessing mitigation targets in relation to fair shares, tracing efforts, a comparison between nations[3,68], and quantifying a country's responsibility to global $CO_2$ removal efforts[47]. These applications underscore the importance of transparent and NGHGI-consistent national RCB calculations, which we seek to facilitate with this study. We stress once more that net zero $CO_2$ emissions are not sufficient for halting warming when using NGHGI accounting conventions for anthropogenic $CO_2$[11]. Hence, recognition of the existence of different RCB definitions (IPCC-based and NGHGI-consistent) is essential to prevent misconceptions when designing climate strategies and accurately assessing the compatibility of national with global climate targets, such as in NDC synthesis reports and future legal cases.

## Methods
### Calculating a NGHGI-consistent global RCB

By default, we consider global RCB estimates for 1.5 °C C (50%) and 2 °C (66%) calculated for the 1 January 2023 by Lamboll et al. ($RCB_{glob}^{Lamboll\,et\,al.}$)[24]. While a RCB only accounts for $CO_2$ emissions, each global temperature limit is associated with a specific amount of future $CO_2$ and non-$CO_2$ climate forcer emissions, derived from an assessed set of future scenarios[58]. Non-$CO_2$ GHG and aerosol emissions are expected to contribute to net warming (yellow bar in Fig. 1a), with their emission pathways implicitly affecting the size of the global IPCC-based RCB (blue bar in Fig. 1a). As we focus on consistent $CO_2$ accounting, we use IPCC-based RCB values as they are published and do not further examine assumptions behind future non-$CO_2$ pathways (see discussion). We calculate IPCC-consistent (also termed IPCC-based) global RCBs ($RCB_{glob}^{IPCC}$) for the 1 January 1990–2024 with total global fossil $CO_2$ emissions ($E_{CO_2}^{foss}$) taken from the Global Carbon Budget 2024 (GCB2024)[30]. To account for the definition of anthropogenic LULUCF $CO_2$ emissions used by IPCC Assessment Reports (ARs) we further add the mean of LULUCF $CO_2$ emissions estimated by four bookkeeping models (BMs, variable: $E_{CO_2}^{LULUCF,BM_b}$, $b \in [1, 4]$) also

provided by the GCB2024[30].

$$
RCB_{glob}^{IPCC}(t) = RCB_{glob}^{Lamboll\,et\,al.}(2023)
$$
$$
+ \begin{cases} \sum_{t'=2023}^{t-1} \left( -E_{CO_2}^{foss}(t') - \frac{1}{4}\sum_{b=1}^{4}\left(E_{CO_2}^{LULUCF,BM_b}(t')\right)\right) & \text{if } t > 2023 \\ \sum_{t'=t}^{2022} \left( E_{CO_2}^{foss}(t') + \frac{1}{4}\sum_{b=1}^{4}\left(E_{CO_2}^{LULUCF,BM_b}(t')\right)\right) & \text{if } t < 2023 \end{cases}
$$

$$(1)$$

To make the global RCB consistent with the NGHGI accounting conventions, we first adjust the global RCB by using the results of the reanalysis performed by Gidden et al.[8]. They quantified the reduction in the global RCB by adopting the definition of anthropogenic LULUCF $CO_2$ emissions implied by the NGHGI accounting conventions from 2020 onward for 1.5 °C (50%) and 2 °C C (66%) scenarios from the AR6 scenario database[69] – this is why we limit our analysis to the RCB for 1.5 °C (50%) and 2 °C (66%). The reduction in global RCB is given by the integrated difference between the estimate of model-based and NGHGI-based anthropogenic LULUCF emissions until net zero $CO_2$ is reached from a model-based perspective and depends on the characteristics of the specific scenario (individual dots in Fig. 1c). The reanalysis by Gidden et al.[8] appears to underestimate the past difference between the (positive) model-based and (negative) NGHGI-based emissions (Supplementary Fig. 2). We therefore adjust their calculated values with data on the difference between mean $CO_2$ emissions from GCB2024 BMs and global NGHGI-reported values[57] from 2020 to 2023, slightly increasing the reduction in the global RCB. This form of adjustment is, however, not possible for the future difference, likely leading to an underestimation of the correction. We use the mean of all reanalyzed C1 (C3) scenarios from the AR6 scenario database[69] for the correction of the 1.5 °C (2 °C) IPCC-based RCB (correction term $\Delta E_{CO_2,\,LULUCF_{IPCC}-LULUCF_{NGHGI}}^{2023\to net\,zero\,CO_2}$ in Eq. (2)).

To estimate the range of future $CO_2$ emissions from bunker fuels we take scenarios from the AR6 scenario database[69], Climate Action

Tracker (CAT) assessments[70,71], historical and scenario data from the International Energy Agency (IEA)[72–74], publicly available scenario data from BloombergNEF[75], and historical bunker fuel $CO_2$ emissions from the GCB2024[30]. In the AR6 scenario database, global $CO_2$ emissions from international aviation and shipping are described by the variables `Emissions|CO2|Energy|Demand|Transportation|Aviation` and `Emissions|CO2|Energy|Demand|Transportation|Maritime`, provided by 8, 7, and 9 scenarios in the C1, C2, and C3 categories, respectively. The IEA Net Zero Scenario[74] and the two BloombergNEF scenarios (Net Zero and Economic Transition) provide data on $CO_2$ emissions from aviation and shipping, separately, but do not discriminate between domestic and international transport. According to IEA data[73], international aviation historically contributed between 58% and 62% of total aviation $CO_2$ emissions. We therefore use a factor of 0.6 to scale provided $CO_2$ emissions from aviation, assuming a near-constant split until net zero $CO_2$ emissions are reached. For shipping, we assume that $CO_2$ emissions are dominated by international transport and use voyage-based data provided by CAT[70], leading to good agreement with the other datasets specifying international shipping and total bunker fuel $CO_2$ emissions in the historical period (Supplementary Fig. 18 and 19). Bunker fuel $CO_2$ emissions are summed from 2023 until the year net zero $CO_2$ is reached to obtain a further correction to the global RCB as of 1 January 2023. In the case of the CAT assessments, we assume net zero $CO_2$ emissions are reached in 2050 (2065) for the 1.5 °C C (2 °C) compatible pathways, following the timing of net zero $CO_2$ emissions found in the other scenarios considered.

The cumulative future $CO_2$ emissions from bunker fuels ($\Delta E_{CO_2, \text{bunker fuels}}^{2023 \to \text{net zero } CO_2}$) are subtracted from the global RCB estimate, already corrected for the difference in the definition of anthropogenic LULUCF $CO_2$ emissions between IPCC ARs and NGHGIs. Given the spread in the correction terms associated with future bunker fuel emissions, and the designedly optimistic values in CAT assessments, we use the mean value of the C1 scenarios for correcting the RCB for 1.5 °C (50%) and the mean value of the C3 scenarios for the 2 °C (66%) global RCB. The full range of estimates is however considered in the illustration of the uncertainty. This way, we obtain a global RCB ($RCB_{\text{glob}}^{\text{NGHGI}}$) that is consistent with the definitions applied in NGHGIs. The conversion of RCBs to the definitions applied in NGHGIs is conceptually shown in Fig. 1a and can be expressed as follows:

$$RCB_{\text{glob}}^{\text{NGHGI}}(2023) =$$
$$RCB_{\text{glob}}^{\text{IPCC}}(2023) - \Delta E_{CO_2, \text{LULUCF}_{\text{IPCC}} - \text{LULUCF}_{\text{NGHGI}}}^{2023 \to \text{net zero } CO_2} - \Delta E_{CO_2, \text{bunker fuels}}^{2023 \to \text{net zero } CO_2} \quad (2)$$

To calculate the NGHGI-consistent global RCB back (forward) to 1 January 1990 (2024), we consider an updated version of the dataset[57] provided by Grassi et al.[31] (available on request) for globally aggregated LULUCF $CO_2$ fluxes reported in NGHGIs ($E_{CO_2}^{\text{LULUCF, NGHGI}}$), and bunker fuel $CO_2$ emissions from the GCB2024 ($E_{CO_2}^{\text{bunker fuels}}$)[30]:

$$RCB_{\text{glob}}^{\text{NGHGI}}(t) = RCB_{\text{glob}}^{\text{NGHGI}}(2023)$$
$$+ \begin{cases} \sum_{t'=2023}^{t-1} \left( -E_{CO_2}^{\text{foss}}(t') + E_{CO_2}^{\text{bunker fuels}}(t') - E_{CO_2}^{\text{LULUCF, NGHGI}}(t') \right) & \text{if } t > 2023 \\ \sum_{t'=t}^{2022} \left( E_{CO_2}^{\text{foss}}(t') - E_{CO_2}^{\text{bunker fuels}}(t') + E_{CO_2}^{\text{LULUCF, NGHGI}}(t') \right) & \text{if } t < 2023 \end{cases}$$
$$(3)$$

## Allocation of the NGHGI-consistent budget to countries

The NGHGI-consistent global RCB is distributed among countries according to different allocation principles, based on past population data from Our World in Data (1850–2023)[76] and supplementary Swiss population data[77], fossil territorial (1850–2023) and consumption-based (1990–2021, see Supplementary Note 3), as well as LULUCF $CO_2$

emissions from the GCB2024[30] and GDP-per-capita data (1950–2022) from the Maddison Project Database[78] estimated using purchasing power parity (using GDP based on market exchange rates was shown to lead to negligible changes in allocations[49]).

The number of countries represented in the datasets varies. National RCBs are calculated for the biggest subset of countries represented in the datasets used, even if some entities are not formally part of the UNFCCC. In addition to the 197 countries that are parties to the UNFCCC, the population dataset includes data for 41 entities that are either only partially recognized countries, disputed territories, or overseas sub-national territories, often with a certain degree of autonomy, which account for around 0.5% of the global population in 2023 (see Section 6 in Supplementary Information for a complete list). The different allocation principles applied are the following, with the number of national RCBs given in parentheses: (cumulative) equal-per-capita (238); Bretschger burden sharing, grandfathering, (cumulative) equal-per-capita with historical responsibility for territorial emissions (all 197), (cumulative) equal-per-capita with historical responsibility for consumption-based emissions (118); capacity and capacity with historical responsibility (both 165).

The mathematical implementation of the different allocation principles is given in the following, where we use:

$$i \ldots \text{Country index}$$
$$\text{Pop}_i(t) \ldots \text{Population} \quad (4)$$
$$\text{GDP}_i^{\text{per-capita}}(t) \cdot \text{Pop}_i(t) = \text{GDP}_i(t) \ldots \text{Gross domestic product}$$

$$\sum_{t'=y}^{t} \text{Pop}_i(t') = \text{cPop}_i^y(t) \ldots \text{Cumulative population since year } y \quad (5)$$

$$\sum_{t'=y}^{t} \text{GDP}_i(t') = \text{cGDP}_i^y(t) \ldots \text{Cumulative GDP since year } y \quad (6)$$

$$E_{CO_2, i}^{\text{foss}}(t) + \frac{1}{4} \sum_{b=1}^{4} \left( E_{CO_2}^{\text{LULUCF, BM}_b}(t) \right) = E_{CO_2, i}^{\text{terr}}(t) \ldots \text{Territorial } CO_2 \text{ emissions} \quad (7)$$

$$\sum_{t'=y}^{t} E_{CO_2, i}^{\text{terr}}(t') = \text{cE}_{CO_2, i}^{\text{terr}, y}(t) \ldots \text{Cumulative territorial } CO_2 \text{ emissions since year } y \quad (8)$$

$$E_{CO_2, i}^{\text{foss, cons}}(t) + \frac{1}{4} \sum_{b=1}^{4} \left( E_{CO_2}^{\text{LULUCF, BM}_b}(t) \right) = E_{CO_2, i}^{\text{cons}}(t) \ldots \text{Consumption} \quad (9)$$
$$- \text{based } CO_2 \text{ emissions}$$

$$\sum_{t'=y}^{t} E_{CO_2, i}^{\text{cons}}(t') = \text{cE}_{CO_2, i}^{\text{cons}, y}(t) \ldots \text{Cumulative consumption} \quad (10)$$
$$- \text{based } CO_2 \text{ emissions since year } y$$

**Equal-per-capita** (EPC): The allocation of the global RCB depends on the current share of the global population.

$$RCB_i^{\text{EPC}}(t) = RCB_{\text{glob}}^{\text{NGHGI}}(t) \cdot \text{Pop}_i(t) \quad (11)$$

**Cumulative equal-per-capita** (cEPC): The allocation of the global RCB depends on the share of the cumulative population since a year $y$.

$$RCB_i^{\text{cEPC}y}(t) = RCB_{\text{glob}}^{\text{NGHGI}}(t) \cdot \text{cPop}_i^y(t) \quad (12)$$

**Bretschger burden sharing:** The fraction of the global RCB allocated to a country depends on a scaled version of the $CO_2$

emissions per person[15]. This represents a variation of the grandfathering principle and, like it, lacks a basis in international environmental law[46]. However, this approach was used to establish exceedance of a national RCB[48]. For allocation of the budget on the 1 January, we take the emissions from the previous year.

$$\text{RCB}_i^{\text{Bretschger}}(t) = \text{RCB}_{\text{glob}}^{\text{NGHGI}}(t) \cdot \frac{m_i(t-1) \cdot F_i(t-1)}{\sum_i (m_i(t-1) \cdot F_i(t-1))} \quad (13)$$

with

$$m_i(t) = \frac{\text{Pop}_i(t)}{\sum_i \text{Pop}_i(t)} \quad (14)$$

$$F_i(t) = \left( \frac{E_{\text{CO}_2,i}^{\text{terr}}(t)}{\text{Pop}_i(t)} \right)^{0.25} \quad (15)$$

**Grandfathering:** The global RCB is distributed according to the fraction of emissions in the year before. Grandfathering is considered to contrast it with other allocation principles. We stress that grandfathering is argued to have no ethical basis in international environmental law[43,46].

$$\text{RCB}_i^{\text{GF}}(t) = \text{RCB}_{\text{glob}}^{\text{NGHGI}}(t) \cdot \frac{E_{\text{CO}_2,i}^{\text{terr}}(t-1)}{\sum_i E_{\text{CO}_2,i}^{\text{terr}}(t-1)} \quad (16)$$

**Capacity** (CAP$y$ ($\sigma = \sigma^*$)): The allocation of the global RCB is proportional to the factor $\phi_{\text{CAP},i}^{y,\sigma^*}$, which depends on a scaled version of the cumulative GDP per capita, with the strength of the scaling encapsulated in the scaling factor $\sigma$[17,18].

$$\text{RCB}_i^{\text{CAP}y(\sigma=\sigma^*)}(t) = \text{RCB}_{\text{glob}}^{\text{NGHGI}}(t) \cdot \frac{\phi_{\text{CAP},i}^{y,\sigma^*}(t)}{\sum_i \phi_{\text{CAP},i}^{y,\sigma^*}(t)} \quad (17)$$

with

$$\phi_{\text{CAP},i}^{y,\sigma^*}(t) = \left( \frac{\text{cGDP}_i^y(t-1)}{\text{cPop}_i^y(t-1)} \right)^{-\sigma^*} \cdot \text{cPop}_i^y(t-1) \quad (18)$$

The choice of $\sigma$ is normative. Following Pelz et al.[17,18] use $\sigma = 0.5$ as a default, but do calculations with $\sigma = 1$ as well.

For the allocation principles and methods so far, historical responsibility can be accounted for by calculating a country's fair share for a past point in time and subtracting the $CO_2$ emissions that occurred since then. The allocation principles and methods described next take historical responsibility explicitly into account.

**Equal-per-capita with historical responsibility for territorial / consumption-based emissions** (EPC+Terr$y$ / EPC+Cons$y$):[4,79]

$$\text{RCB}_i^{\text{EPC}+\text{Terr}y}(t) = \text{RCB}_i^{\text{EPC}}(t) - \sum_{t'=y}^{t-1} \left( E_{\text{CO}_2,i}^{\text{terr}}(t') - \frac{\sum_i E_{\text{CO}_2,i}^{\text{terr}}(t')}{\sum_i \text{Pop}_i(t')} \cdot \text{Pop}_i(t') \right) \quad (19)$$

$$\text{RCB}_i^{\text{EPC}+\text{Cons}y}(t) = \text{RCB}_i^{\text{EPC}}(t) - \sum_{t'=y}^{t-1} \left( E_{\text{CO}_2,i}^{\text{cons}}(t') - \frac{\sum_i E_{\text{CO}_2,i}^{\text{cons}}(t')}{\sum_i \text{Pop}_i(t')} \cdot \text{Pop}_i(t') \right) \quad (20)$$

We note that we do not fully account for consumption-based LULUCF $CO_2$ emissions. Deforestation can be driven by exports[80], but consumption-based LULUCF $CO_2$ emissions have not been systematically quantified to the best of our knowledge.

**Cumulative equal-per-capita with historical responsibility for territorial / consumption-based emissions** (cEPC+Terr$y$ / cEPC+Cons$y$):[47]

$$\text{RCB}_i^{\text{cEPC}+\text{Terr}y}(t) =$$
$$\text{RCB}_i^{\text{cEPC}}(t) - \left( cE_{\text{CO}_2,i}^{\text{terr},y}(t-1) - \frac{\text{cPop}_i^y(t-1)}{\sum_i \text{cPop}_i^y(t-1)} \cdot \sum_i cE_{\text{CO}_2,i}^{\text{terr},y}(t-1) \right) \quad (21)$$

$$\text{RCB}_i^{\text{cEPC}+\text{Cons}y}(t) =$$
$$\text{RCB}_i^{\text{cEPC}}(t) - \left( cE_{\text{CO}_2,i}^{\text{cons},y}(t-1) - \frac{\text{cPop}_i^y(t-1)}{\sum_i \text{cPop}_i^y(t-1)} \cdot \sum_i cE_{\text{CO}_2,i}^{\text{cons},y}(t-1) \right) \quad (22)$$

**Capacity with historical responsibility** (CAPRES$y$ ($\sigma = \sigma^*$)): This allocation principle compares the capacity-based fair-share emissions with actual territorial emissions for each year since the start year of historical responsibility $y$.

$$\text{RCB}_i^{\text{CAPRES}y(\sigma=\sigma^*)}(t) =$$
$$\text{RCB}_i^{\text{cEPC}y}(t) - \sum_{t'=y}^{t-1} \left( E_{\text{CO}_2,i}^{\text{terr}}(t') - \frac{\phi_{\text{CAP},i}^{y,\sigma^*}(t')}{\sum_i \phi_{\text{CAP},i}^{y,\sigma^*}(t')} \cdot \sum_i E_{\text{CO}_2,i}^{\text{terr}}(t') \right) \quad (23)$$

## Global analysis

For global analyses we consider only five allocation methods that take historical responsibility since 1990 into account (EPC+Terr1990, EPC+Cons1990, cEPC+Terr1990, cEPC+Cons1990, CAPRES1990 ($\sigma$=0.5)). We denote the associated national RCBs as "fair-share" RCBs. However, we note that our choice contains an implicit value judgment and may underestimate the full historical responsibility of certain high income countries that disproportionately contributed to global $CO_2$ emissions between 1850 and 1990. Projections until 2035 for national RCB ranges are based on a linear extrapolation of the year-to-year change in the minimum and maximum national RCBs from 2013–2022 (equivalent to a quadratic extrapolation). The same is done for 2000–2022 for sensitivity analysis (Supplementary Tables 1–3). Countries are geographically aggregated, as in the GCB2024[30]. For extrapolation, SSP2 population projections are used and taken from the Wittgenstein Center Population and Human Capital Projections[81].

## Switzerland's remaining carbon budget

Future (cumulative) $CO_2$ emissions for Switzerland are derived from its updated 2025 NDC[56] and the *ZERO Basis* scenario from the Energy Perspectives 2050+[82]. In this scenario, Switzerland's net GHG emissions (aggregated according to GWP100 from the IPCC AR5) reach zero in 2050 and net zero $CO_2$ is reached in 2045 (Supplementary Fig. 20). This scenario is consistent with a GHG emission budget of 106.8 MtCO$_2$-eq for 2031–2035, and reductions in GHG emissions compared to 1990 of 50% reduction in 2030, of 59% over 2031–2035, and of 65% (slightly more than the minimum 64% stated in the NDC) for 2031–2040.

Switzerland's equal-per-capita RCB in 2020 is calculated according to Eq. (11). For recalculating the value communicated in the ruling of the ECtHR, we use the 1.5 °C (67%) global RCB for 1 January 2020 from the IPCC AR6 WG1 Table 5.8[7] of 400 GtCO$_2$. From Table 2 in the Supplementary information of Lamboll et al.[24] we obtain an updated 1.5 °C (66%) RCB estimate for 1 January 2023 of 60 GtCO$_2$ and use Eq. (1) to calculate the value for 1 January 2020 yielding 274 GtCO$_2$. We calculate the NGHGI-consistent values using the available correction

terms for 1.5 °C (50%) (Eq. (2)), which we judge to be very close to the necessary correction for 1.5 °C (66%).

## Reporting summary
Further information on research design is available in the Nature Portfolio Reporting Summary linked to this article.

## Data availability
All data used and generated during this study have been deposited at https://doi.org/10.5281/zenodo.17426185.

## Code availability
All code used to analyze the data and generate the figures of the main text and Supplementary Information is available at https://doi.org/10.5281/zenodo.17426185.

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

## Acknowledgements

Konstantin Weber, Cyril Brunner, and Reto Knutti are part of SPEED2-ZERO, a Joint Initiative co-financed by the ETH Board. We thank Giacomo Grassi and Joana Melo for their valuable support in obtaining the carbon flux data from national greenhouse gas inventories and Gergana Gyuleva for her helpful comments on the first version of the draft.

## Author contributions

C.B. and R.K. initiated and supervised the research. K.W. performed the analysis, created the figures and wrote the first draft. All authors contributed to the continuous discussion of results, further analyses, and editing of the manuscript.

## Funding

## Competing interests

The authors declare no competing interests.

## Additional information

**Peer review information** : *Nature Communications* thanks Michael Gillenwater and the other, anonymous, reviewer(s) for their contribution to the peer review of this work. A peer review file is available.

