## [Transparent Peer Review file · Nature Communications]

Tracking Country-level Mitigation Progress Using NGHGI-Consistent Carbon Budgets

Corresponding Author: Mr Konstantin Weber

Version 0:

Reviewer comments:

Reviewer #1

(Remarks to the Author)
GENERAL COMMENTS

* This is an excellent, and highly thorough, analysis of national carbon budget allocation...raising an excellent and neglected point regarding the construction of national GHG inventory accounting rules versus global modeling. Each makes assumptions and applies boundary rules appropriate to their separate intended uses. But when these uses overlap, problems can arise. This manuscript presents a valuable and novel surfacing of this overlooked issue.

The authors properly highlight, repeatedly, the inherent ethical and normative choices involved in any allocation option. And so present their findings across a range of choices and options.

While the analysis presented in the paper appears sound, my one major critique is that it could do a better job of explaining to the reader the specific differences in emissions/removals accounting between NGHGs and in modeling conventions. Lines 50-70 provide a summary, but the reader is still left wanting a better elaboration. If you are a modeler, you want to understand how specific assumptions you take for granted differ from those used in NGHGs and why. And vice versa for those who are NGHGI experts. Maybe these assumptions should be questioned. Or maybe not...but even then, experts in each community should come away understanding at a technical level the differences between the two scientific practice conventions so they can convey and properly caveat their work across communities of practice. Also, many modelers and NGHGI experts are not steeped in LULCF carbon accounting details. So, I encourage you to expand on this explanation in some manner.

SPECIFIC COMMENTS

Line 27: Should you not refer to maximum "net" CO2 emissions or refer to "emissions and removals"

Line 38: suggest replacing the word "falls", which reads as odd, with the term "undermined"

Line 75-78: Can you inform readers why this difference in passive uptake exists? It will go a long way to helping the reader grasp these differences if rationales are given for why each community of practice has made the assumptions they way they has.

Line 85-88: It would be appropriate to address the fact that under the UNFCCC process, bunker fuels are not really excluded. National inventories report them as memo items (outside of national totals). And they are addressed through IMO and ICAO processes. So, while they are outside of NDCs in most cases, they are not, in theory, just ignored, as is implied by this paragraph.

Line 120-122: It is not clear, then, how non-CO2 radiative forcing is addressed in your analysis.

Line 128-130: In reference to the general comment above, here is a place where some clear explanation of what is different in the two carbon accounting conventions would be helpful.

Figure 1 is a good and useful illustration of the component parts of the analysis.

Line 208, 249, 316: The introduction of consumption-based emission estimates seems to be without a clear explanation. NGHGs do not involve consumption-based methods. The supplemental material does not appear to provide background on where these consumption-based estimates come from. Is the idea that NGHGI values should be reconfigured into consumption-based reports for the purpose of applying RGBs? The overall comment is simply that this needs a better explanation.

Line 338-341: This summary conclusion is powerful. I found the sentence presented here to be a more useful presentation of this important point than the way it is presented in the Abstract.

Lastly, I want to simply commend the authors on an excellent final concluding paragraph of the paper, which does a nice job of providing the bigger picture perspective on how to interpret the results of this study. Well-done.

Reviewer #2

(Remarks to the Author)

Weber and colleagues note that current approaches to estimate country-level fair shares of a remaining carbon budget (RCB) consistent with global climate objectives yield estimates that are not directly comparable with national climate targets. This lack of comparability is due to differences in accounting conventions between national greenhouse gas inventories (NGHGs) and scientific modeling conventions. The authors estimate an NGHGI-consistent 1.5°C RCB and use different published fair share approaches to evaluate country-level NGHGI-consistent fair shares. They conclude with a case study (Switzerland).

The authors make a fair call for accuracy (i.e., by harmonizing RCB estimates with NGHGs) in fair share estimates, drawing on existing work that has highlighted these mismatches and estimated global NGHGI-consistent RCBs (Gidden et al., 2023).

It is, however, unclear whether this correction yields relevant qualitative insights compared to prior fair share literature contributions. The authors present results across a range of published fair share estimates, ensuring numerical accuracy; it is unclear whether this numerical accuracy significantly changes the qualitative insights from the existing fair share literature. Most of the insights that the authors derive from their quantitative assessment, such as the dependence of fair shares on normative allocation choices and the temporal consumption of emissions, are consistent with findings from previous literature.

The authors try to address the relevance of this correction by using Switzerland as a case study. The results of this “fact check” of the ECHR ruling effectively show that a fair remaining carbon budget on a per-capita basis is significantly lower due to the reduction in the global quantity to be shared. Since this is driven by a decrease in the global quantity to share, I would expect the fair share estimates for other countries to be similarly scaled down, with the differences between the different fair share estimates persisting. What we learn from such an exercise is that the fair share approaches continue to remain more important than the correction factor when we derive qualitative insights.

In this paper, the authors do not address key issues facing the fair share literature, for example, the impending exhaustion of a 1.5°C RCB (i.e., what does one do when there is no quantity to allocate?).

I appreciate that the authors have put significant effort into this paper. The manuscript is well-written, and the methods are documented clearly. I agree with the narrow point on accuracy that the authors make in this paper. While this is a relevant point to make, it has been made elsewhere, and I do not think the authors have sufficiently justified the relevance of a new literature contribution when applied in a “fair shares” context. I am sorry for not being more positive in my review at this stage. I hope that some of my comments are helpful while the authors revise the manuscript.

References:

M. J. Gidden, T. Gasser, G. Grassi, N. Forsell, I. Janssens, W. F. Lamb, J. Minx, Z. Nicholls, J. Steinhauser, K. Riahi, Aligning climate scenarios to emissions inventories shifts global benchmarks. *Nature* 624, 102–108 (2023).

Version 1:

Reviewer comments:

Reviewer #1

(Remarks to the Author)

The authors have sufficiently addressed and responded to my review comments. No further comments.

Reviewer #2

(Remarks to the Author)

Thank you for the opportunity to review the revised manuscript.

I acknowledge the relevance of the authors' call for accuracy (and proposed solution) in how land use emissions and international aviation and shipping are accounted for in national fair shares of a remaining carbon budget consistent with global climate goals.

However, I think the authors could have engaged more comprehensively with my concerns over their manuscript's engagement with broader conceptual and pressing issues facing the fair share literature. These concerns include the implications of a zero (or negative) 1.5°C remaining carbon budget and the implications of the normative range of estimates outweighing the implications of their correction. In my opinion, the authors sidestep these concerns by classifying them as beyond the scope and aim of the manuscript or briefly highlighting them in supplementary notes.

The authors also requested references to prior discussions of international aviation and shipping and LULUCF emissions in the fair share literature. I mention two below (and there are many more) that have discussed these issues and either applied corrections (for international aviation and shipping) or discussed why they exclude LULUCF emissions given uncertainties (section: References). This does not reduce the relevance of the authors' correction but hopefully helps them place it in the context of prior discussions on the issue.

References

Rajamani, L., Jeffery, L., Höhne, N., Hans, F., Glass, A., Ganti, G., & Geiges, A. (2021). National 'fair shares' in reducing greenhouse gas emissions within the principled framework of international environmental law. *Climate Policy*, 21(8), 983-1004.

Robiou du Pont, Y., Jeffery, M. L., Gütschow, J., Rogelj, J., Christoff, P., & Meinshausen, M. (2017). Equitable mitigation to achieve the Paris Agreement goals. *Nature Climate Change*, 7(1), 38-43.

Version 2:

Reviewer comments:

Reviewer #2

(Remarks to the Author)

REVIEWER COMMENTS

Reviewer #1 (Remarks to the Author):

GENERAL COMMENTS

* This is an excellent, and highly thorough, analysis of national carbon budget allocation...raising an excellent and neglected point regarding the construction of national GHG inventory accounting rules versus global modeling. Each makes assumptions and applies boundary rules appropriate to their separate intended uses. But when these uses overlap, problems can arise. This manuscript presents a valuable and novel surfacing of this overlooked issue.

Thank you for the interest in our study, the detailed suggestions and your positive review. We appreciate that you took the time to read and critically review our manuscript.

The authors properly highlight, repeatedly, the inherent ethical and normative choices involved in any allocation option. And so present their findings across a range of choices and options.

While the analysis presented in the paper appears sound, my one major critique is that it could do a better job of explaining to the reader the specific differences in emissions/removals accounting between NGHGs and in modeling conventions. Lines 50-70 provide a summary, but the reader is still left wanting a better elaboration. If you are a modeler, you want to understand how specific assumptions you take for granted differ from those used in NGHGs and why. And vice versa for those who are NGHGI experts. Maybe these assumptions should be questioned. Or maybe not...but even then, experts in each community should come away understanding at a technical level the differences between the two scientific practice conventions so they can convey and properly caveat their work across communities of practice. Also, many modelers and NGHGI experts are not steeped in LULCF carbon accounting details. So, I encourage you to expand on this explanation in some manner.

We agree that this is an important point to improve the comprehensibility of the manuscript. We have therefore expanded our explanation in the introduction in the following way (lines 59-70 of the original manuscript and lines 59-97 of the revised manuscript):

Scientific modeling conventions and NGHGI accounting differ in their definitions of anthropogenic CO₂ emissions in two ways: The first difference occurs in the ~~accounting definition~~ of land-use, land-use change, and forestry (LULUCF) CO₂ emissions, ~~as where~~ scientific modeling conventions and NGHGs ~~use different definitions of anthropogenically influenced (managed) land~~^{11,27,28}. ~~Additionally implicitly attribute different parts of atmosphere-land CO₂ fluxes to human influence~~^{11,27-29}. In scientific modeling conventions that underpin the IPCC assessments, indirect CO₂ fluxes – ~~such as those driven by the fluxes driven by human-caused changes to the environment, such as elevated atmospheric CO₂ levels and higher temperatures, higher temperatures, and changes in nutrient supply~~ – are not counted as anthropogenic emissions or removals in IPCC assessments. ~~These indirect~~ CO₂ fluxes, but are considered to be part of the natural land sink¹¹. These indirect effects, also termed “passive”, ~~arise due to effects on vegetation from past emissions rather than a sustained anthropogenic influence. In contrast, in NGHGI accounting,~~ have so far led to a strong net uptake of atmospheric CO₂³⁰. NGHGI accounting^{9,10}, however, relies largely on observational data, making it typically difficult to fully separate passive CO₂ fluxes ~~are not fully separated from direct~~ from CO₂ fluxes due to direct anthropogenic influence (e.g., ~~de- and afforestation) on~~, re-, and afforestation or forest

management). For practical reasons, NGHGI accounting uses land that is classified ~~as managed~~^{27,28} by countries as “managed” as an indicator for the land where anthropogenic CO₂ fluxes occur^{27,28}. Part of what is considered the natural CO₂ sink by modelers is incidentally included as an anthropogenic CO₂ sink in NGHGIs, as countries largest in area tend to classify most (if not all) land as managed and isolation of passive CO₂ fluxes remains imperfect in NGHGI accounting²⁹. This results in a fundamental mismatch of 5–7 GtCO₂ per year, corresponding to 12–17% of 2023’s global total anthropogenic CO₂ emissions^{30,31} ~~that~~. This gap continues to evolve depending on the climate scenario^{8,32} ~~and the land classification by countries~~. This difference in methodology also explains why IPCC reports⁷ and assessments of the Global Carbon Project³⁰ find that the LULUCF sector is a net source of CO₂ emissions, while NGHGIs consistently report it as a net sink^{8,27,28,32,27–29,32}.

~~Crucially, Allen et al.⁴⁴ demonstrate that~~ Crucially, under scientific modeling conventions, the concept of reaching net zero CO₂ to stop global temperature rise works just because passive CO₂ fluxes are classified to be natural (non-anthropogenic)¹¹. After reaching net zero CO₂, continued passive CO₂ uptake by the land and oceans leads to decreasing atmospheric CO₂ concentrations that are necessary to counterbalance continued warming after forcing stabilization³³. On the contrary, under NGHGI accounting of anthropogenic LULUCF CO₂ emissions, achieving net zero anthropogenic CO₂ emissions does not halt global temperature rise. ~~The passive uptake of CO₂ counted as a natural (non-anthropogenic) sink in scientific modeling conventions is necessary to counterbalance residual warming after net zero CO₂ is reached³³. However,~~ as demonstrated by Allen et al.¹¹. Most of the passive CO₂ uptake ~~is partly classified as anthropogenic in NGHGIs and increasing pressure to re-classify passive uptake as anthropogenic deteriorates this issue⁴⁴. Hence, the IPCC-based RCB continues to deplete after occurring on managed land is regarded as anthropogenic CO₂ removal in NGHGIs²⁹ and can therefore be potentially used to compensate for anthropogenic CO₂ emissions¹¹. Thus, after reaching global net zero CO₂ emissions~~ ~~are reached under NGHGI accounting conventions, requiring net negative CO₂ emissions to limit~~ from a NGHGI accounting perspective, the IPCC-based RCB continues to deplete. Limiting warming to a set temperature threshold ~~–This requires continuous net negative CO₂ emissions in NGHGI accounting – this makes the concept of a RCB fundamentally incompatible with NGHGI accounting. Correcting the size of the available global RCB to fit NGHGI accounting conventions therefore requires assumptions about future emission pathways⁸.~~

SPECIFIC COMMENTS

Line 27: Should you not refer to maximum "net" CO₂ emissions or refer to "emissions and removals"

Thank you for spotting this. As you suggested, we added the word “net” in the revised manuscript (lines 29-31):

They represent a country’s share^{4,5} of the global RCB – the maximum net CO₂ emissions permissible before exceeding a specific temperature threshold with a given probability^{6,7} – under certain assumptions and value judgments.

Line 38: suggest replacing the word "falls", which reads as odd, with the term "undermined"

Agreed. We adapted the sentence accordingly (lines 39-42 of the revised manuscript):

Conceptually, this mismatch cannot be entirely eliminated, because the *scientific foundation of the global RCB falls-is undermined* when using the reporting guidelines^{9,10} used in national greenhouse gas inventories (NGHGI), as additional global warming does not stop when reaching net zero CO₂ emissions¹¹.

Line 75-78: Can you inform readers why this difference in passive uptake exists? It will go a long way to helping the reader grasp these differences if rationales are given for why each community of practice has made the assumptions they way they has.

Thank you for this suggestion. We tried to give a more complete explanation of why models and NGHGs differ in their consideration of passive uptake in lines 63-75 of the revised manuscript.

In scientific modeling conventions that underpin the IPCC assessments, indirect CO₂ fluxes – such as those driven by the fluxes driven by human-caused changes to the environment, such as elevated atmospheric CO₂ levels and higher temperatures, higher temperatures, and changes in nutrient supply – are not counted as anthropogenic emissions or removals in IPCC assessments. These indirect CO₂ fluxes, but are considered to be part of the natural land sink¹¹. These indirect effects, also termed “passive”, arise due to effects on vegetation from past emissions rather than a sustained anthropogenic influence. In contrast, in NGHGI accounting, have so far led to a strong net uptake of atmospheric CO₂³⁰. NGHGI accounting^{9,10}, however, relies largely on observational data, making it typically difficult to fully separate passive CO₂ fluxes are not fully separated from direct from CO₂ fluxes due to direct anthropogenic influence (e.g., de- and afforestation) on-, re-, and afforestation or forest management). For practical reasons, NGHGI accounting uses land that is classified as managed^{27,28}–by countries as “managed” as an indicator for the land where anthropogenic CO₂ fluxes occur^{27,28}. Part of what is considered the natural CO₂ sink by modelers is incidentally included as an anthropogenic CO₂ sink in NGHGs, as countries largest in area tend to classify most (if not all) land as managed and isolation of passive CO₂ fluxes remains imperfect in NGHGI accounting²⁹.

We hope that this gives at least some insight to members of both communities. A more detailed explanation can be found in the scientific publications we cite.

Line 85-88: It would be appropriate to address the fact that under the UNFCCC process, bunker fuels are not really excluded. National inventories report them as memo items (outside of national totals). And they are addressed through IMO and ICAO processes. So, while they are outside of NDCs in most cases, they are not, in theory, just ignored, as is implied by this paragraph.

This is an important clarification to make, as we did not mean to imply that these emissions are ignored, but most often (as you correctly point out) not within the scope of NDCs. The corresponding paragraph has been adapted and now reads as follows (lines 85-88 of original manuscript and lines 99-106 of revised manuscript):

The second difference relates to CO₂ emissions from international aviation and shipping (bunker fuels), as NGHGs report but exclude these emissions. Under the United Nations Framework Convention on Climate Change (UNFCCC), their mitigation is coordinated in cooperation with the International Civil Aviation Organization (ICAO) and the International Maritime Organization (IMO)³⁴. NGHGs still report CO₂ emissions from bunker fuels but exclude them from national totals^{99,35}. Similarly, bunker fuel emissions are most often excluded from national climate targets; with only a few exceptions (e.g.,. Exceptions include intra-EU aviation emissions³⁶, Switzerland’s

net zero⁹⁷ and UK climate targets⁹⁸, intra-EU aviation emissions⁹⁶ target³⁷, and the UK climate target³⁸.

Line 120-122: It is not clear, then, how non-CO₂ radiative forcing is addressed in your analysis.

Thank you for this important feedback. We explain this in our Methods (this part was removed and adapted from the beginning of the Results – lines 118-123 of the original manuscript, now lines 444-451 of the revised manuscript):

While a RCB only accounts for CO₂ emissions, each global temperature limit is associated with a specific amount of future CO₂ and non-CO₂ climate forcer emissions, derived from an assessed set of future scenarios⁴⁸. Non-CO₂ GHG and aerosol emissions are expected to contribute to net warming (yellow bar in Fig. 1a), with their emission pathways implicitly affecting the size of the global IPCC-based RCB (blue bar in Fig. 1a). As we focus on consistent CO₂ accounting, we use IPCC-based RCB values as they are published and do not further examine assumptions behind future non-CO₂ pathways (see discussion).

The relevant paragraph in our discussion reads as follows (lines 346-355 of the original manuscript; lines 364-373 of the revised manuscript):

Besides uncertainties in applied correction terms, NGHGI-consistent RCB estimates inherently contain geophysical and scenario-related uncertainties from IPCC-based RCB estimates^{2,6,24}, particularly regarding future non-CO₂ emission pathways^{6,48}. Less stringent non-CO₂ GHG mitigation could further reduce global RCB estimates by approximately 220 GtCO₂⁷, implying depletion of the 1.5 °C (50%) NGHGI-consistent global RCB already around 2021. While integrating non-CO₂ GHGs into a broader global warming budget⁵⁰ would offer a more comprehensive picture of climate responsibility, such an approach adds conceptual and methodological complexities. Given the conceptual simplicity, robust scientific foundation, and longstanding acceptance of carbon budgets⁵⁸⁻⁶¹, we focus here on CO₂ for introducing NGHGI-consistent RCBs.

Line 128-130: In reference to the general comment above, here is a place where some clear explanation of what is different in the two carbon accounting conventions would be helpful.

Thank you for this suggestion! We decided to give a bit more background in the Introduction (see our response to your comment of lines 75-78 above and lines 59-81 of the revised manuscript). We hope that this clarifies the most important differences.

Figure 1 is a good and useful illustration of the component parts of the analysis.

Thank you!

Line 208, 249, 316: The introduction of consumption-based emission estimates seems to be without a clear explanation. NGHGIs do not involve consumption-based methods. The supplemental material does not appear to provide background on where these consumption-based estimates come from. Is the idea that NGHGI values should be reconfigured into consumption-based reports for the purpose of applying RGBs? The overall comment is simply that this needs a better explanation.

This is a very good comment and an interesting question. We added a Supplementary Note (Supplementary Note 3) that gives a bit of context. Lines 32-49 of the revised Supplementary Information now read as follows:

Supplementary Note 3: The use of consumption-based emissions

The IPCC guidelines for national greenhouse gas inventories specify that “National inventories include greenhouse gas emissions and removals taking place within national territory and offshore areas over which the country has jurisdiction”⁸, making clear that under the UNFCCC process reporting is based on territorial emissions. Likewise, national climate strategies and NDCs exclusively cover territorial emissions. An alternative perspective is provided by consumption-based accounting, which adjusts territorial CO₂ emissions for the carbon intensity of imported and exported goods (data provided, for example, in the Global Carbon Budget 2024⁹). This approach has been discussed in the literature as a way of attributing emissions to countries in line with their consumption patterns, and thus as a potential basis for assessing historical responsibility^{10–12}. Some scholars have argued that it better captures national responsibility for climate change (e.g.,¹³). Even some countries’ updated NDC¹⁴ mentions the necessity to consider consumption-based emissions in its reflection on fair shares. Calculating national RCBs that account for historical responsibility for consumption-based CO₂ emissions is therefore a complementary (and arguably necessary) perspective, even though they may not be directly comparable to NDCs, and NGHGs continue to be based on territorial emissions.

In our study, we take consumption-based CO₂ emissions from the Global Carbon Budget 2024, as written in our Methods (lines 513-519 of the revised manuscript):

The NGHGI-consistent global RCB is distributed among countries according to different allocation principles, based on past population data from Our World in Data (1850–2023)⁷⁵ and supplementary Swiss population data⁷⁶, fossil territorial (1850–2023) and consumption-based (1990–2021, see Supplementary Note 3), as well as LULUCF CO₂ emissions from the GCB2024³⁰ and GDP-per-capita data (1950–2022) from the Maddison Project Database⁷⁷ estimated using purchasing power parity (using GDP based on market exchange rates was shown to lead to negligible changes in allocations⁴⁸).

Line 338-341: This summary conclusion is powerful. I found the sentence presented here to be a more useful presentation of this important point than the way it is presented in the Abstract.

Thank you for this suggestion. We adapted the abstract (also in response to the comments of the other reviewer and keep its length limited), which now reads (lines 7-24 of the revised manuscript):

*The remaining carbon budget (RCB) of individual countries provides a benchmark to evaluate national mitigation efforts and was ~~recently~~ central to a ~~prominent court~~ recent European Court of Human Rights’ ruling. However, consistent estimates of national RCBs are complicated by mismatches in the accounting methodology of anthropogenic CO₂ between scientific studies and national greenhouse gas inventories (NGHGI). Here, we **address a methodological inconsistency in the mentioned ruling and quantify how the alignment with NGHGI accounting reduces the global RCB. ~~We calculate~~** For 2024, alignment with NGHGs reduces the 1.5 °C (50%) global RCB by 50% (~100 GtCO₂) and the 2 °C (66%) global RCB by 20% (~200 GtCO₂). We anticipate that permissible country-reported CO₂ emissions compatible with the 1.5 °C limit, when aggregated globally, have either already been exhausted or will reach zero by 2028. To facilitate consistent application of national RCBs in the future, we provide a dataset of NGHGI-consistent*

national RCBs for a wide range of allocation methods and countries, illustrating that national RCBs inherently depend on normative choices. Interpreting Paris Agreement equity principles, we find large global inequalities and a lack of progress towards achieving the agreed targets. By that by 2025, 64–85 countries (representing ~50% of global GDP) have exceeded their fair-share RCB for 1.5 °C (50% likelihood). We also identify a methodological inconsistency in the European Court of Human Rights' ruling involving Switzerland and the KlimaSeniorinnen and provide methodologically robust RCBs for future applications. Even if national RCBs are unlikely to directly drive global climate negotiations due to their dependence on normative choices, our framework enables NGHGI-consistent methodologically more robust RCB calculations to accurately track and assess country-level mitigation progress.

Lastly, I want to simply commend the authors on an excellent final concluding paragraph of the paper, which does a nice job of providing the bigger picture perspective on how to interpret the results of this study. Well-done.

Simply: Thank you!

Reviewer #2 (Remarks to the Author):

Weber and colleagues note that current approaches to estimate country-level fair shares of a remaining carbon budget (RCB) consistent with global climate objectives yield estimates that are not directly comparable with national climate targets. This lack of comparability is due to differences in accounting conventions between national greenhouse gas inventories (NGHGs) and scientific modeling conventions. The authors estimate an NGHGI-consistent 1.5°C RCB and use different published fair share approaches to evaluate country-level NGHGI-consistent fair shares. They conclude with a case study (Switzerland).

Thank you for taking the time to critically review our manuscript and providing valuable feedback! We hope to address your points raised and clarify our contribution to the scientific literature.

The authors make a fair call for accuracy (i.e., by harmonizing RCB estimates with NGHGs) in fair share estimates, drawing on existing work that has highlighted these mismatches and estimated global NGHGI-consistent RCBs (Gidden et al., 2023).

You are right to point out our aim for better accuracy. We also agree that some of the issues have been raised before, particularly regarding differences in the attribution of passive CO₂ fluxes in the LULUCF sector. However, we emphasize that we are not aware of a study where both necessary corrections – also including bunker fuels – have been systematically applied in the context of global and national RCBs. The work by Gidden et al. (2023) addresses one part of this correction, without quantifying or updating the global RCB or national RCBs. Apart from a single document that focuses on one country and a single allocation approach (Robiou du Pont & Nicholls, 2023), we have not found literature that additionally considers future CO₂ emissions from bunker fuels. Only by considering both corrections, we can enable a more consistent comparison between national RCBs and NGHGs / NDCs. To the best of our knowledge, the concept of an NGHGI-consistent global RCB, that takes into account either one or both corrections, has also not been introduced in the scientific literature.

References:

Gidden, Matthew J., et al. "Aligning climate scenarios to emissions inventories shifts global benchmarks." *Nature* 624.7990 (2023): 102-108.

Robiou Du Pont, Yann. & Nicholls, Zebedee. "Calculation of an emissions budget for Switzerland based on Bretschger's (2012) methodology" tech. rep. (2023), 14.

https://www.klimasenioren.ch/wp-content/uploads/2023/04/230427_53600_20_Annex_Doc_2_Robiou_du_Pont_Nicholls_Expert_Report.pdf.

Moreover, while the derivation of national RCBs from an adjusted global RCB may be methodologically straightforward for informed experts (using published fair-share allocation approaches), it needs to be done in each individual case and with a transparent methodology. We think that accurate and transparent national RCB calculations are highly relevant, given that the argumentation of the ECHR may serve as a precedent for further legal cases. This is especially true given the recent advisory opinion of the International Court of Justice (ICJ) on the "Obligations of States in respect of climate change." Hence, we provide a dataset with the study that does contain NGHGI-consistent national RCBs back to 1990 for all 197 Parties to the UNFCCC. This will make it easier for a broader spectrum of users to compare national RCBs with national climate targets or NDCs, as well as to compare different countries.

To better highlight our contribution to the literature, we have adapted and expanded the end of our introduction (lines 124-136 in the revised manuscript):

~~*This study presents a methodology for approximate national RCBs consistent with NGHGI accounting conventions, providing results for and thereby strengthen the robustness of national RCBs, when used to evaluate country-level mitigation progress. To simplify such a procedure, we provide a dataset of NGHGI-consistent national RCBs for a wide range of allocation methods and for all 197 countries that are parties to the UNFCCC based on common allocation principles and methods. We aim to clarify subtle differences in the attribution of passive CO₂ accounting relevant to the quantification of (national) RCBs for the assessment of national climate targets, including the third round of NDCs due in 2025 (which most countries have yet to submit). By incorporating a temporal perspective, we uncover how well national CO₂ emissions have historically aligned with the goals of the Paris Agreement. Finally, we examine Switzerland's RCB, subject to discussion in the ruling of the ECHR. fluxes in the LULUCF sector have been documented^{8,32}, and the correction for bunker fuels has been done once before⁴⁷, to our knowledge, the two corrections have so far not been applied systematically to the global and national RCBs. We aim to fill a gap in the scientific literature concerning an up-to-date, methodologically more robust quantification of national RCBs that combines global scope, temporal coverage, and a broad range of allocation methods – also provided for any users within a single dataset. Here, we quantify the effect of the proposed correction and also examine the variation in the updated national RCBs that arises from different normative choices.*~~

It is, however, unclear whether this correction yields relevant qualitative insights compared to prior fair share literature contributions. The authors present results across a range of published fair share estimates, ensuring numerical accuracy; it is unclear whether this numerical accuracy significantly changes the qualitative insights from the existing fair share literature. Most of the insights that the authors derive from their quantitative assessment, such as the dependence of fair shares on normative allocation choices and the temporal consumption of emissions, are consistent with findings from previous literature.

Thank you for raising this point. As you rightly point out, the correction we apply ensures increased accuracy in the quantification of national RCBs, and this is our primary aim with this manuscript. However, in our original manuscript, we did not communicate clearly enough how the proposed correction affects national RCBs and how it compares to differences in allocation approaches. As you correctly point out in your next comment, we find that the choice of the allocation approach tends to have a bigger effect on national RCBs than the correction we propose. To contextualize and highlight this, we have added the following to our revised manuscript (lines 200-205 of the revised manuscript):

While alignment with NGHGI accounting substantially changes the size of the national RCB for some allocation principles – e.g., China’s 2022 EPC-based RCB is reduced by 39%, or its 2022 CAPRES1990 ($\sigma=1$)-based RCB is almost completely depleted – in terms of magnitude, the choice between the different allocation principles often has a larger effect on the resulting national RCB (see Supplementary Fig. 4 and 5 for the effect of the correction on national RCBs).

We have also added Supplementary Fig. 4 and 5 to illustrate the relationship between IPCC-based and NGHGI-consistent national RCBs. For national RCBs that do not take historical responsibility into account, the correction just scales down all national RCBs by the same factor (and as long as the distributable budget is positive, they stay positive, of course). But for allocation approaches that include historical responsibility, there are instances where the IPCC-based RCB is positive, while the NGHGI-based RCB is negative. This relationship depends on the size of the correction we apply (which we judge to be rather on the low side, see Supplementary Note 1), the temperature limit (we show the relationship for 1.5 °C (50%), but the change is bigger when considering 2 °C (66%)), and the year the global RCB is distributed. Thus, there are instances where the correction changes the sign of the national RCB and instances where it does not matter. If the correction turns out to be bigger than what we estimate, there may be many more cases where national RCBs changes sign, and the proposed conceptual framework may still prove useful.

We therefore believe that there are interesting qualitative insights. Since the global budget to share is substantially smaller (by up to hundreds of GtCO₂), the assessment of whether or when a country exceeds its “fair share” of the remaining carbon budget according to a certain allocation approach may change. In other words, not considering the difference between different CO₂ accounting conventions leads to a systematic overestimation of the distributable budget and hence also a systematic overestimation of national budgets in the previous literature. We believe that this should be addressed and corrected, hence our proposed correction. To clarify this matter, we include the following two aspects in our manuscript:

1. We find a sizable decrease in the 1.5 °C and 2 °C-compatible remaining carbon budgets that can be distributed to countries, and hence earlier depletion of the GCB when adopting CO₂ accounting conventions used in NGHGs and NDCs (lines 351-362 of the revised manuscript):

The RCB distributable to countries is lower than the IPCC-based RCB ~~, though the and not considering alignment with NGHGs leads to a systematic overestimation of national RCBs, skewing assessments of national mitigation efforts.~~ The exact correction terms bear considerable uncertainty due to scenario dependence. ~~Furthermore, they~~ They are likely to be underestimated (Supplementary Note 1) and the NGHGI-consistent global RCB ~~is thus still thus still remains~~ overestimated. Consequently, we anticipate that permissible country-reported CO₂ emissions compatible with the 1.5 °C limit, when

aggregated globally, have either already been exhausted or will reach zero by 2028 – slightly earlier than estimated ~~than~~ in the most recent NDC synthesis report²². The 2024 NGHGI-consistent 2 °C (66%)-compatible global RCB amounts to approximately 700 GtCO₂, equivalent to just 21 years of country-reported CO₂ emissions from 2022^{30,57}, emphasizing the need for rapid global decarbonization to stay within the Paris Agreement temperature limits.

2. There is a shift to earlier years when assessing when a country has exceeded its “fair share” of a global remaining carbon budget. We updated Fig. 4 of the original manuscript in our revised manuscript, which now contrasts the fraction of countries / population / GDP exceeding their “fair-share” RCB when derived from an NGHGI-consistent vs. an IPCC-based global RCB (see also Supplementary Fig. 9, 10, 16, 17). To highlight this, we have added the following to our revised manuscript (lines 282-286 of the revised manuscript):

Without alignment to NGHGIs, exceedance shifts to later years, as indicated in gray in Fig. 4 and Supplementary Fig. 10: For example, when considering IPCC-based RCBs in 2022, the number of countries exceeding their 2 °C (66%)-compatible RCB, the population they represent and their share of global GDP are reduced to 29–48 countries, covering 0.9–1.4 billion people, and 30–43% of global GDP, respectively.

This can be compared to the results in the case with alignment to NGHGIs, which we state in lines 276-279 of the revised manuscript:

We find a similar result for 2 °C (66% ~~likelihood~~): 37–57 countries out of 197 (covering 1.2–1.6 billion people and 37–47% of global GDP) had already surpassed their fair share of CO₂ emissions by 2022 when using NGHGI-consistent RCBs (Supplementary Fig. ~~8~~ 10 and Supplementary Tables 1–3).

Subsequently, we also have added a comparison with the IPCC-based estimate in the discussion (lines 399-403 of the revised manuscript):

Even with this conservative choice, we find that at the time of the Paris Agreement negotiations (end of 2015), ~~42–73~~ 49–69 countries had already exceeded their 1.5 °C-compatible fair share of the global NGHGI-consistent RCB – compared to 44–62 (a bias of around 10%) when using an IPCC-based RCB.

While discussions about fair shares are not the main aim of our work, we think that it is important to highlight the updated differences in national RCBs that exist due to normative choices and give some context. To be clearer that dependence on normative choices is widely known, we have adapted our manuscript as follows (lines 238-242 of the revised manuscript):

*In brief, the selection of countries in Fig. 2 highlights the **widely recognized** strong dependence of national RCBs on both normative allocation choices and implementation details: ~~These~~^{13,16,17,21,43,51}. Here we find that **these** tend to dominate over the **correction applied and the** sensitivity to the chosen temperature target (see Supplementary Fig. ~~4 for 2-C (66%) for 7~~ for comparison with Fig. 2).*

To be clear about our focus on the methodological correction that is necessary for alignment with NGHGs, we have adapted our abstract – also in response to the comments of another reviewer (lines 7-24 of the revised manuscript):

The remaining carbon budget (RCB) of individual countries provides a benchmark to evaluate national mitigation efforts and was ~~recently~~ central to a ~~prominent court~~ recent European Court of Human Rights' ruling. However, consistent estimates of national RCBs are complicated by mismatches in the accounting methodology of anthropogenic CO₂ between scientific studies and national greenhouse gas inventories (NGHGI). Here, we ~~address a methodological inconsistency in the mentioned ruling and~~ quantify how the alignment with NGHGI accounting reduces the global RCB. ~~We calculate~~ For 2024, alignment with NGHGs reduces the 1.5 °C (50%) global RCB by 50% (~100 GtCO₂) and the 2 °C (66%) global RCB by 20% (~200 GtCO₂). We anticipate that permissible country-reported CO₂ emissions compatible with the 1.5 °C limit, when aggregated globally, have either already been exhausted or will reach zero by 2028. To facilitate consistent application of national RCBs in the future, we provide a dataset of NGHGI-consistent national RCBs for a wide range of allocation methods and countries, ~~illustrating that national RCBs inherently depend on normative choices~~. Interpreting Paris Agreement equity principles, we find ~~large global inequalities and a lack of progress towards achieving the agreed targets. By that by 2025, 64–85 countries (representing ~50% of global GDP) have exceeded their fair-share RCB for 1.5 °C (50% likelihood). We also identify a methodological inconsistency in the European Court of Human Rights' ruling involving Switzerland and the KlimaSeniorinnen and provide methodologically robust RCBs for future applications.~~). Even if national RCBs are unlikely to directly drive global climate negotiations ~~due to their dependence on normative choices~~, our framework enables ~~NGHGI-consistent~~ methodologically more robust RCB calculations to ~~accurately track and~~ assess country-level mitigation progress.

For clarification, we have changed lines 175-178 of the original manuscript (lines 188-192 of the revised manuscript):

We do not attempt to judge the different allocation assumptions ~~here~~ and their implications in this work, but rather to correct them for consistency with the NGHGI methodology and provide them to any potential user. Many other allocation criteria have been proposed and are possible, ~~but we argue that the ones presented provide a good overview~~ while here we provide a number of allocation principles commonly found in the literature.

We have also shortened part of the results sections that describes temporal changes in national RCBs (lines 231-250 of original manuscript, now lines 256-260 of the revised manuscript):

~~We observe the following: Several countries depleted their fair share RCBs early. For instance, the US per capita RCB for National RCB depletion over time differs widely (Fig. 3a, b): The US 1.5 °C (50%) C-compatible RCB turned negative already around the year 2000 across all five selected RCB allocation methods. Despite a decreasing annual depletion rate, the US continues to exceed and continues to decline faster than the global average rate of per capita RCB reduction (dashed black line in Fig. 3) – this highlights its role as a disproportionately high CO₂ emitter. Factors related to (economic) development appear to influence the rate of RCB depletion (a metric related to a country's CO₂ emissions): Whereas, China's per capita RCB has remained positive and slightly above RCB depletes more rapidly than the global average, it has declined at an accelerating rate since the late 2000s. Until around 2005, China's RCB trajectory closely mirrored that of Nigeria. However, their depletion rates diverged considerably thereafter, and by 2022, China since around 2010, and Switzerland's per capita RCB declined~~

~~much faster than the global average as a result of rapid economic growth. For some countries, the pronounced sensitivity to the allocation method leads to a broad RCB range, complicating a conclusive assessment: Across the five allocation methods considered, Switzerland's RCB spans both positive and negative values over multiple decades without a distinct time of RCB depletion. While the upper bound of the range closely mimics the global trend, allocation methods that incorporate historical responsibility for consumption-based emissions persistently yield a much lower RCB and faster depletion. RCB spans positive to negative values depending on the chosen allocation method over a long period of time.~~

The authors try to address the relevance of this correction by using Switzerland as a case study. The results of this “fact check” of the ECHR ruling effectively show that a fair remaining carbon budget on a per-capita basis is significantly lower due to the reduction in the global quantity to be shared. Since this is driven by a decrease in the global quantity to share, I would expect the fair share estimates for other countries to be similarly scaled down, with the differences between the different fair share estimates persisting. What we learn from such an exercise is that the fair share approaches continue to remain more important than the correction factor when we derive qualitative insights.

You are correct in pointing out that the differences between the different fair share estimates persist and that for many countries our proposed correction translates to a change that is less important than the choice of the allocation approach chosen (see our comments above, in particular lines 200-205 of the revised manuscript and the new Supplementary Fig. 4 and 5). While it is true that the choice of the specific allocation approach can be a major determinant of the size of a national RCB, this does not diminish the importance of the correction we apply to be consistent with NGHGI accounting conventions. Our correction can be conceptualized as a methodological bias correction that affects all national RCBs regardless of the chosen allocation approach. This bias correction is evidence-based and grounded in the best available science. By contrast, the choice of an allocation approach is inherently normative and, in practice, political. These represent two entirely different dimensions of uncertainty when quantifying national RCBs. Here, we improve upon the methodological dimension and provide transparency regarding the normative dimension and nuances in its operationalization in allocation approaches

Furthermore, the magnitude of corrections to the IPCC-based global RCB may be larger than some of the presented results suggest. We focus primarily on 1.5 °C-compatible RCBs in the 2020s, however, the magnitude of corrections gains in importance (i) for higher temperature limits and increasing delay in achieving net zero CO₂, (ii) earlier points in time (e.g., Fig. 1d and Supplementary Fig. 3), and (iii) when accounting for a potential underestimation of the correction terms (see Supplementary Note 1). For example, aligning the 2 °C (66%) global RCB in 1990 with NGHGI accounting conventions may reduce it by more than 500 GtCO₂ (> 85 tCO₂ capita⁻¹); a shift that is large enough to drastically affect assessments based on national RCBs.

In other words, which allocation approaches are used in the end in the legal or political context is a political choice. We can make sure that national RCBs are calculated as robustly as possible and estimates based on the best available scientific understanding are readily available to whoever wishes to use them.

There are also instances when a single allocation approach is chosen and used for argumentation, for example in legal settings (such as by the ECHR). Under these circumstances, it is less important that different fair share estimates differ, and the methodologically robust quantification gains in importance. In addition, as national RCB estimates appear in policy

documents and legal deliberations, we strongly believe that we should strive for methodological rigor in calculating national RCB estimates.

We also pointed out in our response to your comment above that the allocation approach chosen can also have a stronger influence on national RCB values than the chosen temperature limit (lines 238-242 of the revised manuscript). Still, the distinction between RCBs for 1.5 °C (50%) or 2 °C (66%) is arguably important.

Finally, previous refinements of the global RCB have been recognized as important contributions to the literature because they improved the robustness of the global budget. To place our proposed correction into context, we compare it with the last three successive updates of the global RCB: (1) the Special Report on Global Warming of 1.5 °C (Rogelj et al., 2018), (2) the update in IPCC AR6 WGI Chapter 5 (Canadell et al., 2021), and (3) the most recent refinement by Lamboll et al., 2023. Using historical CO₂ emissions from the Global Carbon Budget 2024 (Friedlingstein et al., 2024), we calculate global RCBs for 2024 to cancel out the effect of emissions since 2024 and find that the AR6 update changed the 1.5 °C (50%) and 2 °C (66%) RCBs by ~0 GtCO₂ (0%) and ~ +60 GtCO₂ (+7%), respectively, compared to Rogelj et al., 2018. The update in Lamboll et al. (2023) led to a further change of ~ -130 GtCO₂ (-39%) and ~ -90 GtCO₂ (-9%) for the 2024 global RCBs compared to AR6. By comparison, aligning RCBs with NGHGI accounting conventions results in reductions of ~ -100 GtCO₂ (-47%) for 1.5 °C (50%) and ~ -200 GtCO₂ (-21%) for 2 °C (66%) in 2024. Thus, the magnitude of our proposed correction falls within, or even exceeds, the normal range of previous updates to the global RCB. There is evidence that such updates are valuable to the scientific community, since future work builds on previous methodological updates, where the discussion and study of fair shares happens in parallel.

References:

Rogelj, J., D. Shindell, K. Jiang, S. Fifita, P. Forster, V. Ginzburg, C. Handa, H. Kheshgi, S. Kobayashi, E. Kriegler, L. Mundaca, R. Séférian, and M.V. Vilariño, 2018: Mitigation Pathways Compatible with 1.5°C in the Context of Sustainable Development. In: *Global Warming of 1.5°C. An IPCC Special Report on the impacts of global warming of 1.5°C above pre-industrial levels and related global greenhouse gas emission pathways, in the context of strengthening the global response to the threat of climate change, sustainable development, and efforts to eradicate poverty* [Masson-Delmotte, V., P. Zhai, H.-O. Pörtner, D. Roberts, J. Skea, P.R. Shukla, A. Pirani, W. Moufouma-Okia, C. Péan, R. Pidcock, S. Connors, J.B.R. Matthews, Y. Chen, X. Zhou, M.I. Gomis, E. Lonnoy, T. Maycock, M. Tignor, and T. Waterfield (eds.)]. Cambridge University Press, Cambridge, UK and New York, NY, USA, pp. 93-174. <https://doi.org/10.1017/9781009157940.004>.

Canadell, J.G., P.M.S. Monteiro, M.H. Costa, L. Cotrim da Cunha, P.M. Cox, A.V. Eliseev, S. Henson, M. Ishii, S. Jaccard, C. Koven, A. Lohila, P.K. Patra, S. Piao, J. Rogelj, S. Syampungani, S. Zaehle, and K. Zickfeld, 2021: Global Carbon and other Biogeochemical Cycles and Feedbacks. In *Climate Change 2021: The Physical Science Basis. Contribution of Working Group I to the Sixth Assessment Report of the Intergovernmental Panel on Climate Change* [Masson-Delmotte, V., P. Zhai, A. Pirani, S.L. Connors, C. Péan, S. Berger, N. Caud, Y. Chen, L. Goldfarb, M.I. Gomis, M. Huang, K. Leitzell, E. Lonnoy, J.B.R. Matthews, T.K. Maycock, T. Waterfield, O. Yelekçi, R. Yu, and B. Zhou (eds.)]. Cambridge University Press, Cambridge, United Kingdom and New York, NY, USA, pp. 673–816, doi:10.1017/9781009157896.007.

Lamboll, Robin D., et al. "Assessing the size and uncertainty of remaining carbon budgets." *Nature Climate Change* 13.12 (2023): 1360-1367.

Friedlingstein, Pierre, et al. "Global carbon budget 2024." *Earth System Science Data Discussions* 2024 (2024): 1-133.

In this paper, the authors do not address key issues facing the fair share literature, for example, the impending exhaustion of a 1.5°C RCB (i.e., what does one do when there is no quantity to allocate?).

While we agree that this is an extremely relevant issue, and while we decided to include a short paragraph in our discussion (lines 406-416 of the revised manuscript) we feel that a detailed discussion would go beyond the scope and aim of our manuscript:

Even when the global RCB for 1.5 °C or 2 °C is depleted, the quantification of national RCBs remains informative, e.g., to estimate the size of a national carbon debt and the resulting required cumulative amount of net negative CO₂ emissions for any internationally agreed temperature limits^{48,62}. Although a detailed discussion of these conditions is outside the scope of this work, it is worth highlighting that under an overshoot of a global RCB, care needs to be taken when applying existing allocation methods, as interpretations of distributional approaches change when moving from a positive to a negative quantity to distribute (Supplementary Note 2). Countries with minimal or negative RCBs thus need not only to adopt ~~but also implement ambitious~~ and implement emission reduction strategies ~~, aiming for~~ with the highest possible ambition^{1,63}, but also net CO₂ removal after they reach net zero CO₂ ~~is reached~~.

We also added a short Supplementary Note to the Supplementary Information (lines 22-31 of the revised Supplementary Information):

*Supplementary Note 2: Change of interpretation when allocating a negative global RCB
Some allocation approaches assign a larger share of the global RCB to countries with higher emissions (e.g., grandfathering and Bretschger burden sharing). While the global RCB remains positive, this benefits high-emitting countries. By contrast, capacity-based approaches reduce the allocated share with increasing GDP per capita, reducing the share of the global RCB for wealthier nations. However, once a global temperature limit is overshoot and the global RCB becomes negative, these interpretations invert: Approaches that previously penalized wealthy countries now advantage them, and vice versa. Caution is therefore required when applying the same operationalization principles once the global RCB switches sign.*

We also note that the alignment with NGHGs remains relevant after exhaustion of a global RCB.

I appreciate that the authors have put significant effort into this paper. The manuscript is well-written, and the methods are documented clearly. I agree with the narrow point on accuracy that the authors make in this paper. While this is a relevant point to make, it has been made elsewhere, and I do not think the authors have sufficiently justified the relevance of a new literature contribution when applied in a "fair shares" context. I am sorry for not being more positive in my review at this stage. I hope that some of my comments are helpful while the authors revise the manuscript.

Thank you for the feedback and comments. We agree that the way we framed some aspects in our original manuscript was not always aligned with what we wanted to convey. Your comments have been very helpful for rethinking the presentation and framing of our results.

To summarize, we think that (i) the point of accuracy is not as narrow as presented in our original manuscript, (ii) we are not aware of any other contribution to the literature, where the differences in CO₂ accounting, its implications and the necessary two corrections are discussed and are applied systematically, in particular in the context of national RCBs, and (iii) our main conclusions aim not to extend the literature on “fair shares”, but that of the methodology behind global and national RCB, which is relevant whenever national RCBs are calculated for comparison with national climate targets, for example in the legal context. We still want to provide transparency on the normative dimension, even though our results are just consistent with previous literature.

As for the second point, if we have missed any literature that applies the two corrections to the global or national RCB, we would appreciate it if you would share it with us, as we find the topic most relevant.

We sincerely thank you for your comments and hope that we could address them in a satisfying manner. Furthermore, we believe that through your input, we could substantially improve our manuscript.

References:

M. J. Gidden, T. Gasser, G. Grassi, N. Forsell, I. Janssens, W. F. Lamb, J. Minx, Z. Nicholls, J. Steinhauser, K. Riahi, Aligning climate scenarios to emissions inventories shifts global benchmarks. *Nature* 624, 102–108 (2023).

We would like to thank both reviewers and the editor for commenting on our revised manuscript. Please find our replies to all the comments and suggestions highlighted in blue. *New sections of text from the revised manuscript are in blue italics, unchanged sections from the original manuscript are in gray italics, and deleted sections from the original manuscript are struck through in red italics.*

REVIEWER COMMENTS

Reviewer #1 (Remarks to the Author):

The authors have sufficiently addressed and responded to my review comments. No further comments.

We thank the reviewer for their positive assessment.

Reviewer #2 (Remarks to the Author):

Thank you for the opportunity to review the revised manuscript.

I acknowledge the relevance of the authors' call for accuracy (and proposed solution) in how land use emissions and international aviation and shipping are accounted for in national fair shares of a remaining carbon budget consistent with global climate goals.

However, I think the authors could have engaged more comprehensively with my concerns over their manuscript's engagement with broader conceptual and pressing issues facing the fair share literature. These concerns include the implications of a zero (or negative) 1.5°C remaining carbon budget and the implications of the normative range of estimates outweighing the implications of their correction. In my opinion, the authors sidestep these concerns by classifying them as beyond the scope and aim of the manuscript or briefly highlighting them in supplementary notes.

We appreciate the reviewer's continued engagement and their acknowledgment of the relevance of our corrections.

We understand the reviewer's concern regarding the broader conceptual challenges in the fair-share literature. However, our view remains that the perceived issue of a zero or negative global remaining carbon budget (RCB) in the context of fair shares is largely beyond the scope of our manuscript. We acknowledge that the reviewer would prefer a more detailed discussion, but our focus remains methodological in nature, addressing the so far lacking consistency between the global remaining carbon budget (RCB) and the CO₂ emissions reported by countries in their national greenhouse gas inventories (NGHGs). With this, we aim to provide a methodologically more robust quantification of the RCB, both on a global level and for common allocation principles on the national level.

We emphasize that we did not intend to sidestep the reviewer's earlier comments. On the contrary, we accommodated the reviewer's comments by incorporating additional points in

our results and discussion section, and, as correctly pointed out, in the Supplementary Information. As the raised issues are important, but not central points of our work, we consider it appropriate to have certain details and new illustrations in the Supplementary Information. We also provided an extensive explanation of why we are convinced that the correction applied has significance, even when normative choices dominate for a number of countries. In contrast to the reviewer's comment, the normative range of estimates is sometimes, but by no means always, bigger than the quantitative implications of the applied correction. As argued in the previous response, even in cases where the normative range of estimates is indeed bigger, this does not warrant ignoring the correction. We remain convinced that our current framing appropriately presents the scope of the manuscript and adequately situates our contribution within the wider literature.

Still, in response to this and the reviewer's earlier comment, we have previously added and now adapted a paragraph in our discussion illustrating our view on the conditions when the global RCB for 1.5 °C may be depleted (lines 406-416 in the previous version of the manuscript and lines 410-422 in the updated version of the manuscript):

Even when the global RCB for 1.5 °C or 2 °C is depleted, the quantification of national RCBs remains informative, e.g., to estimate the size of a national carbon debt and the resulting required cumulative amount of net negative CO₂ emissions for any internationally agreed temperature limits^{48,62}. ~~Although a detailed discussion of these conditions is outside the scope of this work, it is~~ When historical responsibility is taken into account, many national RCBs are already negative today and increasingly so once the global RCB is depleted. Conversely, certain countries may keep positive budgets long after a global RCB is depleted. It is worth highlighting that under an overshoot of a global RCB, care needs to be taken when applying existing allocation methods, as interpretations of distributional approaches change when moving from a positive to a negative quantity to distribute (Supplementary Note 2). ~~Countries~~ However, there is no change in the notion that countries with minimal or negative RCBs ~~thus~~ need not only to adopt and implement emission reduction strategies with the highest possible ambition^{1,63}, but also ~~net aim for net negative CO₂ removal emissions~~ after they reach net zero CO₂.

Finally, we recall that our goal is to provide scientifically robust national RCBs when a normative choice has already been made, to facilitate their application, for example, in the legal context. The goal is to enable national RCB calculations such that they are consistent with what they are usually compared to. We mention the legal context, as national remaining carbon budgets have been and will be (trusting the words of climate litigation experts) employed in attempts to establish a country's exceedance of its fair share of global CO₂ emissions (e.g., Robiou du Pont and Nicholls, 2023). While we see the task of the fair-share literature mainly in informing allocation principles and methods, we aim to provide the corresponding global RCB with the best possible methodological consistency.

The authors also requested references to prior discussions of international aviation and shipping and LULUCF emissions in the fair share literature. I mention two below (and there are many more) that have discussed these issues and either applied corrections (for international aviation and shipping) or discussed why they exclude LULUCF emissions given uncertainties (section: References). This does not reduce the relevance of the authors' correction but hopefully helps them place it in the context of prior discussions on the issue.

We thank the reviewer for providing these references. We note, however, that these studies fundamentally differ from what we aim to do: The provided studies (and others) exclude LULUCF emissions, while we make an adjustment to ensure consistency with national reporting under NGHGs. Also, the provided studies do not discuss this issue of consistency with NGHGs, as was brought forward by Grassi et al. (2018). In our previous response, we were referring to missing work that ensures such consistency (*“To the best of our knowledge, the concept of an NGHGI-consistent global RCB, that takes into account either one or both corrections, has also not been introduced in the scientific literature.”*).

For additional context, consider the following:

Based on the accounting methodology from the IPCC Assessments, which is also used to quantify the Global Remaining Carbon Budget, global anthropogenic CO₂ emissions from 2014 to 2023 were approximately 40 GtCO₂ / year, of which about 4 GtCO₂ / year originated from the LULUCF sector (e.g., Friedlingstein et al., 2025; Grassi et al., 2025). In contrast, aggregated country-reported global CO₂ emissions based on NGHGI methodology are 33 GtCO₂ / year in the same period, including LULUCF emissions that sum to roughly -3 GtCO₂ / year (Grassi et al., 2025). Subsequently, just subtracting LULUCF emissions from country-reported global CO₂ emissions yields about 36 GtCO₂ / year, which are just not the same as global emissions from IPCC Assessments (40 GtCO₂ / year) or as globally aggregated country-reported emissions (33 GtCO₂ / year). Further insights are nicely summarized in Grassi et al. (2025).

Hence, our approach, by contrast to the provided studies, *corrects* the global RCB so that anthropogenic LULUCF CO₂ emissions are treated consistently with NGHGI conventions, and we highlight this point repeatedly and throughout our introduction. This enables the derivation of national RCBs that are directly comparable to national emissions reports and NDCs. Furthermore, we investigate the effect of bunker fuel emissions more systematically and with a transparent methodology.

In response to the reviewer’s comment and to acknowledge the approaches in previous literature, we have added the following to the end of our introduction in our revised manuscript lines 124-136 in the previous version of the manuscript and lines 124-140 in the updated version of the manuscript):

In this study, we propose a correction of the distributable global RCB that improves the consistency with NGHGI accounting, and thereby strengthen the robustness of national RCBs, when used to evaluate country-level mitigation progress. To simplify such a procedure, we provide a dataset of NGHGI-consistent national RCBs for a wide range of allocation methods and for all 197 Parties to the UNFCCC. Previous studies have sometimes excluded CO₂ emissions from the LULUCF sector and bunker fuels when allocating emissions or carbon budgets^{12,46,47}, and accordingly made adjustments to distributable emissions. However, these studies did not address the conceptual mismatch between scientific modeling conventions and NGHGs. While the implications of differences in the attribution of passive CO₂ fluxes in the LULUCF sector have been documented^{8,32}, and ~~the correction for bunker fuels has been done once before~~ a correction has been included once before in the context of a national RCB⁴⁸, to our knowledge, the two necessary corrections have so far not been applied systematically to the global and national

RCBs. We aim to fill a gap in the scientific literature concerning an up-to-date, methodologically more robust quantification of national RCBs that combines global scope, temporal coverage, and a broad range of allocation methods – also provided for any users within a single dataset. Here, we quantify the effect of the proposed correction and also examine the variation in the updated national RCBs that arises from different normative choices.

References:

Robiou Du Pont, Yann. & Nicholls, Zebedee. "Calculation of an emissions budget for Switzerland based on Bretschger's (2012) methodology" tech. rep. (2023), 14.

https://www.klimaseniorinnen.ch/wp-content/uploads/2023/04/230427_53600_20_Annex_Doc_2_Robiou_du_Pont_Nicholls_Expert_Report.pdf.

Grassi, Giacomo, et al. "Reconciling global-model estimates and country reporting of anthropogenic forest CO₂ sinks." *Nature Climate Change* 8.10 (2018): 914-920.

Friedlingstein, Pierre, et al. "Global carbon budget 2024." *Earth System Science Data Discussions* 2024 (2024): 1-133.

Grassi, Giacomo, et al. "Improving land-use emission estimates under the Paris Agreement." *Nature Sustainability* (2025): 1-3.

References

Rajamani, L., Jeffery, L., Höhne, N., Hans, F., Glass, A., Ganti, G., & Geiges, A. (2021). National 'fair shares' in reducing greenhouse gas emissions within the principled framework of international environmental law. *Climate Policy*, 21(8), 983-1004.

Robiou du Pont, Y., Jeffery, M. L., Gütschow, J., Rogelj, J., Christoff, P., & Meinshausen, M. (2017). Equitable mitigation to achieve the Paris Agreement goals. *Nature Climate Change*, 7(1), 38-43.